# Sustained decline in tobacco purchasing in Denmark during the COVID-19 pandemic

Toke R. Fosgaard[1], Alice Pizzo[2] & Sally Sadoff[3] ✉

**Abstract**

**Background** An estimated 8 million people die every year due to tobacco use. The COVID-19 pandemic has increased the health consequences of smoking, which is a leading risk factor for more severe COVID-19 symptoms, hospitalization, and death. The pandemic has also led to reductions in physical activity, increases in stress and declines in mental well-being, all factors commonly associated with triggering higher tobacco use.

**Methods** Using a longitudinal data set of purchasing behavior from 2019–2020 among a national sample of the Danish population ($n = 4042$), we estimate changes in tobacco use during the COVID-19 pandemic. Our analysis compares tobacco purchases prior to the pandemic to purchases during the pandemic, at the individual level. We also examine effects within subgroups based on smoking behavior in 2019 prior to the pandemic. We estimate effects for smokers and non-smokers and, within smokers, for occasional smokers and regular smokers.

**Results** We find large, sustained decreases in tobacco purchases during COVID-19. We estimate that weekly tobacco purchase rates decline by 24% and average quantities decline by 12% during the period spanning the onset of the pandemic in March 2020 through the end of the year. The declines are driven by regular smokers with little change in behavior among nonsmokers and increases in purchases among occasional smokers. Among regular smokers, purchase rates decline by about 30%, tobacco purchases decline by about 20% and quitting rates increase by about 10 percentage points.

**Conclusions** Our results suggest that the COVID-19 pandemic could lead to sustained reductions in smoking.

**Plain language summary**

Being a smoker makes it more likely that a person will have severe COVID-19 symptoms, enter hospital, or die. This study assessed the amount of tobacco products purchased each week during 2019–2020 by a sample of the Danish population. The rate of people purchasing cigarettes weekly went down by 24% and the average quantities purchased decreased by 12% between March 2020 and the end of 2020. Regular smokers purchased about 30% less often and they purchased about 20% fewer cigarettes. Regular smokers were more likely to quit smoking than in the previous year. Our results suggest that the pandemic could lead to long term reductions in smoking.

[1] University of Copenhagen, Department of Food and Resource Economics, Copenhagen, Denmark. [2] Copenhagen Business School, Department of Management, Society and Communication, Copenhagen, Denmark. [3] University of California San Diego, Rady School of Management, San Diego, CA, USA. ✉email: ssadoff@ucsd.edu

Tobacco consumption is one of the leading causes of death globally[1], and one of the most difficult health behaviors to change. An estimated 8 million people die every year due to tobacco use, which causes 17 percent of deaths in the U.S., 16 percent in Europe and 12 percent globally[1]. Tobacco products are highly addictive with extremely low rates of cessation[2]. While tobacco use has declined in the last decade, about 20 percent of people around the world still smoke, including 14 percent in the U.S. and 28 percent in Europe[1,3]. The economic costs of smoking – both health expenditures and productivity losses – are estimated at $1.4 trillion per year globally, with estimates of $300 billion per year in the U.S. and over $600 billion per year in Europe[4,5].

The health risks of smoking have increased during the COVID-19 pandemic, which represents one of the greatest global shocks in modern history to both health and society. Researchers recognized early in the pandemic that tobacco smoking – which is long known to increase susceptibility to infection and activation of inflammation – is a leading risk factor for more severe COVID-19 symptoms, hospitalization, and death[6–8]. The increased health risks were accompanied by unprecedented societal measures including lockdowns, social distancing, and remote working policies. There is evidence from prior work that both health shocks that increase personal risks from smoking (e.g., during pregnancy), as well as large societal shifts (e.g., smoking norms and restrictions) can drive sustained reductions in smoking[9–11]. At the same time, recent studies demonstrate that the pandemic has led to reductions in physical activity, increases in stress, and declines in mental well-being[12–14], all factors commonly associated with triggering higher tobacco use[15]. Taken together, the COVID-19 pandemic could increase, decrease, or have little impact on smoking, and the effects could differ across individuals.

In this study, we examine the impact of COVID-19 on tobacco consumption as proxied by purchases among a national sample of the Danish population ($N = 4042$). We take advantage of a longitudinal data set that tracks individual purchasing behavior prior to and during the pandemic from 2019 through 2020. The purchasing data come from the e-receipt system of widespread use in Denmark, which automatically collects all purchases from the most common supermarkets in the country using individual payment card data (see Materials and Methods for details of the data set).

A rapidly emerging literature has used surveys to investigate self-reported consumption changes during the COVID-19 pandemic for diet and addictive substance use (i.e., alcohol, tobacco, drugs). The survey studies find mixed results for estimates of changes in (expected) tobacco consumption during COVID-19. Some studies report average increases in smoking[16–20], while others report average decreases[21,22]. Within these studies, several find evidence of heterogeneity with about 15–30 percent of respondents reporting increases in smoking and a similar share reporting decreases[19,22,23]. These studies were conducted in different countries (U.S., France, Netherlands, Greece, China, South Africa, India, Italy, the UK), using a range of survey techniques (including online and phone surveys), study lengths (ranging from a week to months), and sample sizes (from a few hundred to many thousands). The only prior study in Denmark was conducted among pregnant women and finds no effects of COVID-19 on smoking among this group[24]. All of these studies rely on self-reported measures. And, as highlighted in a recent study by the U.S. Centers for Disease Control (CDC), they may also suffer from differential response rates during the COVID-19 pandemic compared to pre-pandemic periods[25].

We contribute by estimating the impact of COVID-19 on smoking using systematic purchase data collected at the individual level for the same sample prior to and during the pandemic. Studies using US sales data report 13–14% increases in cigarette sales above expected sales[26] and prior year sales during COVID-19[27]. At the same time, the U.S. CDC estimates declines in the percentage of Americans who report they are current smokers in 2020 compared to 2019 (19.0% down from 20.8%)[25]. The prior studies do not track individual behavior across the comparison periods.

Our study is the first to use individual-level data on tobacco purchases prior to and during the pandemic to examine responses to the COVID-19 pandemic and the associated restrictions. In our context, this allows us to control for individual time-invariant characteristics and to examine heterogeneous individual-level effects that may be masked in aggregated national level estimates. In particular, we characterize our sample based on individual cigarette purchase behavior in 2019 prior to the pandemic: people who do not purchase and people who purchase cigarettes in the pre-pandemic period. We further classify individuals who purchased cigarettes by their purchasing behavior as occasional smokers or regular smokers. We then estimate the impact of COVID-19 among these subgroups to understand whether the pandemic induced non-smokers into smoking, affected occasional smokers whose behavior may be particularly sensitive to social distancing measures, or shifted the habits of regular smokers who are the most likely to be addicted to cigarettes.

We find that weekly cigarette purchase rates decline by 24% and average quantities decline by 12% between March 2020 and the end of the year. Among regular smokers, purchase rates decline by about 30% and purchase quantities by about 20%. Moreover, regular smokers are about 10 percentage points more likely to quit smoking during COVID-19 compared to the same period in the previous year. Our results suggest that the pandemic could lead to long term reductions in smoking.

## Methods

The sample of this study is composed of 4042 users of two smartphone apps developed by *Spenderlog*, a Danish fintech company. The anonymous purchase data was purchased from the private company partner following the European GDPR regulation in force at the time of the study. Informed consent was not required for this study as the data collecting company de-identified the data before transfer. The cleaned dataset includes users who have data for both 2019 and 2020 (Stata/MP 16.1 was used for data analysis). The apps provide an overview of either spending or the carbon footprint of users' grocery purchases (of the 69.4% of respondents we have data on app usage: 48.5% used the spending app, 40.9% used the environmental impact app and 10.6% used both).

There was no recruitment for this study, as individuals were not aware that their cigarette purchase was being observed, and no compensation was involved. Users would typically find out about these apps through marketing campaigns run by the developer company. App-users can activate a profile that includes optional demographic questions and connect the app to an e-receipt system of widespread use in Denmark. The e-receipt system collects data from the most common supermarkets in the country using individual payment card data and registers all food purchases at the individual level without the need for any manual entries. It also provides historic data on grocery purchases prior to the download, allowing us to track purchases within an individual user over time.

A subsample of our participants self-reported demographic characteristics via the app during the sign-up process: gender (68% report), age (68.5% report), employment status (70.4% report), household type (71.5% report). As shown in Supplementary Table S1, compared to the overall population, our participants are on average more likely to be female, to cover younger age groups, to have a higher income, to have children in the household and to be more heavily drawn from the capital region (Copenhagen).

Our analysis focuses on cigarette purchases, which account for 98.67% of tobacco purchases. Including purchases of rolling tobacco does not affect the results (Supplementary Table S2). Our data include the date of each purchase and the number of cigarette packages purchased if any – in Denmark, cigarettes are only available in packages of 20 cigarettes. We aggregate all purchases made in a given week and analyze weekly cigarette purchase rates and quantities. Weekly cigarette purchase rates are the share of individuals who purchased at least one package of cigarettes in a given week. We measure weekly cigarette quantity as the total number of cigarettes purchased in a week, which is equivalent to 20 times the number of packages purchased.

We characterize each individual based on cigarette purchasing behavior in 2019 prior to the COVID-19 pandemic: nonsmokers who purchased no cigarettes in 2019 (76.7% of our sample) and smokers who purchased at least one cigarette in 2019 (24.3%). Among smokers ($n = 983$), we also characterize regular smokers as those who purchased the equivalent of at least one cigarette per day in 2019 ($n = 325$, 33% of smokers, 8% of our sample) and occasional smokers who purchased at least one cigarette in 2019 but less than the equivalent of one cigarette per day ($n = 658$, 66.9% of smokers, 16.3% of our sample). According to the National Institute for Health Education Risk Prevention (INPES), a regular smoker is somebody who admits to smoking at least one cigarette (or equivalent) per day[28]. To calculate average daily cigarette purchases, we divide an individual's total cigarette purchases in 2019 by 365. Our data do not allow us to use another common survey measure of current smoking: having ever smoked 100 or more cigarettes within one's lifetime and smoking every day or some days at the time of survey[25].

The proportion of regular smokers in our data is lower than in the national Danish population: an estimated 17% of Danes smoke daily[29]. To address the non-representativeness of our sample, we include a re-weighted analysis that uses inverse probability weighting to match the shares of smokers and non-smokers observed in the national population, and the results do not change (Supplementary Table S3). The differences between our sample and the Danish population may be in part due to individuals purchasing cigarettes at outlets other than grocery stores that are not captured in our data, such as convenience stores and corner shops. Therefore, our estimates of cigarette purchases may be an underestimate of total quantity. On the other hand, purchases may also reflect household rather than individual tobacco usage and so may be an overestimate of individual consumption. We may also overestimate consumption if people store the cigarettes they buy for long periods. In our data, most consumers buy cigarettes in small quantities suggesting they are not stocking up. We discuss potential concerns related to the measurement of purchases in more detail in the Validity section.

We also examine alcohol purchases, which may serve as a substitute[30] or complement[31] for cigarettes (Supplementary Table S4). We are not able to estimate substitution to e-cigarettes, cigarette smoking cessation products, prescription medication or illegal drugs because we do not observe purchases of these products in our dataset.

**Statistics and reproducibility**. Our primary analysis examines weekly purchases from 2019 through 2020. We focus on the following periods of interest: weeks 1–10 of 2020 prior to the pandemic, weeks 11–20 (March 11th to May 10th, 2020) at the onset of the pandemic during the strict lockdown in Denmark, and the remainder of the year, weeks 21–52. We compare the pandemic period in 2020 to the same period in 2019 using a within-subject comparison. We estimate changes in purchases during COVID-19 using the following regression with individual fixed effects (Supplementary Table S5):

$$y_{it} = \alpha_i + \beta_1\sigma + \beta_2\gamma + \beta_3\gamma\sigma + \beta_4\log(\varpi_t) + \beta_5 x_1 + \beta_6 x_2 + \beta_7 x_3 + \varepsilon_{it}$$

$$(1)$$

where the dependent variable $y_{it}$ of Eq. (1) above is the outcome of interest for individual $i$ at the weekly level $t$; the dummy $\sigma$ is an indicator for the year 2020 (vs. 2019); the dummy $\gamma$ is the time indicator for weeks 11–52; three covariates $x_k$ control for seasonal fixed effects – where $k = 1$ corresponds to spring (week 12–week 25), $k = 2$ to summer (week 26–week 38) and $k = 3$ to fall (week 39–week 52). The individual fixed effect is represented by $\alpha_i$. Our coefficient of interest, $\beta_3$, estimates the term interacting the indicator for 2020 with the indicator for the weeks 11–52, which is the COVID-19 period (in the tables we label the coefficient "COVID-19 period"). We include the logarithmic function of $\varpi_t$ to control for average weekly price as measured in our data; finally, $\varepsilon_{it}$ is the residual term. We cluster standard errors at the individual level. We also separately estimate the impact of COVID-19 during the strict lockdown period and the post-lockdown period through the end of the year (Supplementary Table S6). Because of the structure of the days in a week in 2019 and 2020, week 53 exists only for 2020 and is excluded from the analysis.

## Results

Figure 1 plots average cigarette purchase rates (a) and quantities (b) by week in 2019 and 2020 among individuals who buy at least one cigarette in 2019 (24.3% of our sample). The vertical lines indicate the start of the strict lockdown in response to the onset of the COVID-19 pandemic in Denmark on March 11th, 2020, and the end of the lockdown on May 10th, 2020. During the lockdown (weeks 11–20), non-essential workers in the public sector were ordered to work from home, while in the private sector it was highly recommended; no more than ten people could assemble in public; all indoor public institutions and businesses with close contact were closed; restaurants were closed to in-person dining and could only serve take-out. Stores where cigarettes can be purchased remained open with social distancing regulations in place, which lasted throughout the remainder of the year. After the lockdown ended, occupancy restrictions and social distancing measures remained in place for most businesses. A second, less comprehensive, lock-down was imposed from December 9th, 2020 (week 50) in 38 of the 98 Danish municipalities, including Copenhagen.

Prior to the lockdown period (week 1–10), 2019 and 2020 purchasing habits are almost identical, with individuals buying on average 11.44 cigarettes per week in 2019 and 12.03 cigarettes per week in 2020 ($p = 0.196$ from a rank-sum test of equality at the week level). Purchase rates decline immediately after the lockdown begins, averaging 16.83% during the lockdown compared to 20.59% in the same period in 2019 ($p = 0.002$). However, there is no impact on quantity during this period ($p = 0.931$). After the lockdown ends in 2020, both purchase rates and quantities declined steadily throughout the spring and summer – a period when smoking usually increases – and then level off in the fall. In the post-lockdown period, purchase rates average 13.73% in 2020 compared to 20.68% in the same period in 2019 ($p < 0.0001$), a 34% decline. Quantities over this period average 9.83 cigarettes per week, a 32% decline compared to the 2019 averages of 14.40 ($p < 0.0001$).

Figure 2 plots seasonally adjusted regression coefficients estimating the impact of the pandemic on purchases, pooling the lockdown and post-lockdown periods, weeks 11–52 (see Materials and Methods for details of the regression analysis). We present estimates for the full sample and for subgroups based on

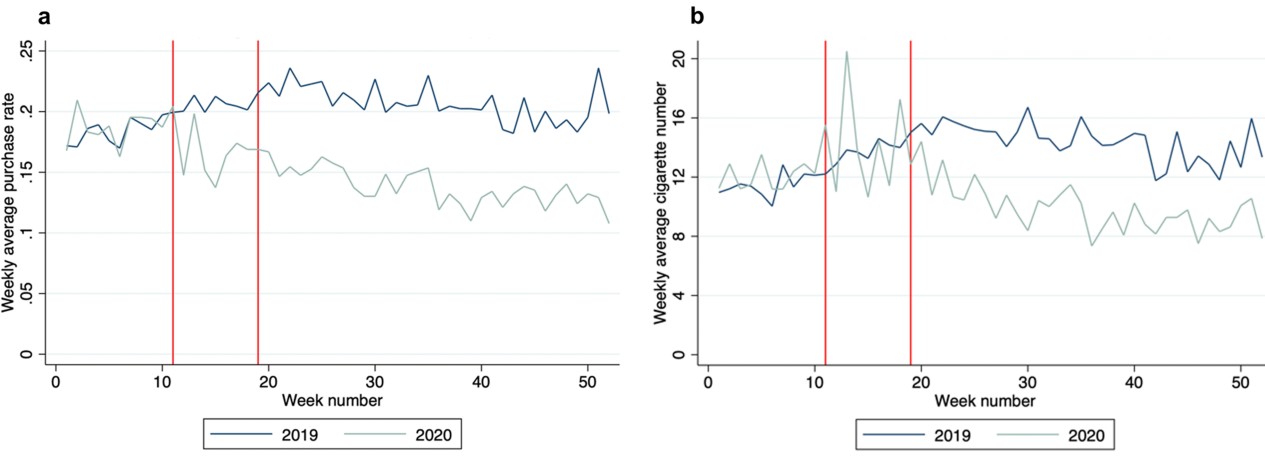

**Fig. 1 Weekly Cigarette Purchases in 2019 vs. 2020. a** Weekly average purchase rate by year; **b** Weekly average purchase of cigarettes by year. The two red lines indicate the lockdown period. The figure only includes individuals who purchased at least one cigarette in 2019. $N = 4042$.

**Fig. 2 Estimated Effects of COVID-19 on Cigarette Purchases for Smokers and Non-Smokers. a** Estimated impact of COVID-19 on weekly purchase rates of cigarettes; **b** Estimated impact of COVID-19 on weekly consumption of cigarettes; **c** Estimated impact of COVID-19 on weekly purchase rates of alcohol; **d** Estimated impact of COVID-19 on weekly consumption of alcohol. All panels include estimates first without and then with correcting for cigarette prices (estimates controlling price are indicated by "p" in parentheses). All coefficients are estimated with individual fixed effects and error terms clustered at the individual level. Error bars shows the 95% confidence interval. "All" $N = 4042$; "Non-smokers" $N = 3059$; "Smokers" $N = 983$; "Occasionals" $N = 658$; "Regulars" $N = 325$.

2019 smoking behavior, both with and without price controls. The panels a and b of Fig. 2 show the estimated impact on weekly cigarette purchase rates (a) and cigarette quantity (b). The full regressions are presented in Supplementary Tables S4–S8.

For the full sample (4042 individuals), we estimate that weekly purchase rates of cigarettes decreased by 1.2–1.6 percentage points ($p < 0.001$ with and without controlling for price), a 24–33% decline compared to the same period in 2019. Weekly quantities drop by an estimated 0.4–1.0 cigarettes ($p = 0.034$ and $p < 0.001$ with and without controlling for price), a 12–28% decline. The declines are driven entirely by smokers ($n = 983$) with no meaningful effects for nonsmokers ($n = 3059$). That is, we find no evidence that the pandemic led non-smokers to begin smoking.

Among the 983 smokers in our sample (who bought at least one cigarette in 2019), we estimate declines of 5.3–6.6 percentage points in weekly purchase rates ($p < 0.001$ with and without controlling for price) and 1.9–4.1 of cigarettes per week ($p = 0.013$ and $p < 0.001$ with and without controlling for price), 26–33% and 14–30% declines respectively. We discuss the effect of cigarette price in more detail in the Mechanisms subsection. We further divide smokers into regular smokers (33%, $n = 325$) and occasional smokers (67%, $n = 658$), where we define regular smokers as those who purchase the equivalent of at least one cigarette per day in 2019[28]. Among regular smokers, we estimate that weekly purchase rates decline by 14.9–18.3 percentage points, a 30–37% percent decline ($p < 0.001$). Average quantities drop by an estimated 7.4–13.4 cigarettes per week, a 20–35% decrease compared to 38.1 cigarettes per week in the same period in 2019 ($p < 0.001$). For occasional smokers – those who purchase at least one cigarette but less than one per day – we find small decreases in purchase rates of 0.5–0.8 percentage points ($p = 0.387$ and $p = 0.187$ with and without controlling for price); and increases in purchase quantities of 0.54–0.83 cigarettes per week ($p = 0.005$ and $p = 0.062$). The decrease in purchase rates but increase in purchase quantities suggests that occasional smokers are buying cigarettes less often but making larger purchases when they do buy. This could be indicative of changes in smoking behavior in this group – for example, a shift away from smoking in social situations to smoking at home.

These results suggest that the overall reduction in cigarette purchases among all smokers is driven by changes among regular smokers. At the same time, the overall decline in cigarette purchases masks heterogeneous effects of the pandemic, with evidence that occasional smokers increased their purchase quantities during COVID-19. We find little evidence of significant heterogeneity across demographic subgroups, including gender, age, household type and employment status (Supplementary Fig. S1).

There may be a concern that smokers substitute reduced tobacco use with increased alcohol consumption. The panels c and d of Fig. 2 present estimates for the impact of the pandemic on alcohol purchase rates (c) and quantities (d). We find suggestive evidence of increases in alcohol purchases during the pandemic. As we discuss in more details below, the increase in alcohol purchases at grocery stores could partially be driven by reduced consumption at restaurants, which our data do not include. Importantly, we find no evidence of larger increases in alcohol purchases among regular smokers, who drive the decrease in cigarette purchases.

Finally, we attempt to estimate quitting behavior among regular smokers in 2020 compared to regular smokers in 2019. We define regular smokers as buying on average at least one cigarette per day in the first ten weeks of the relevant year – i.e., prior to the start of the pandemic period in 2020. Additionally, we define quitting as making no cigarette purchases for the remainder of the year. Figure 3 plots the difference by week in the cumulative proportion of quitters in 2020 compared to 2019 – i.e., the proportion who have made their final cigarette purchase of the year. Quitting rates are significantly higher throughout 2020 compared to 2019. If we consider those who quit for at least six months (i.e., make their final purchase prior to July 1st, week 27), we estimate that quitting rates increase by about 10 percentage points in 2020.

**Validity**. Because our data include grocery store purchases, there may be concern that our estimates could partially reflect changes in where people make purchases. For example, there may be concern that changes in food or alcohol purchases partially reflect a shift towards buying a larger share of food and alcohol at grocery stores during the pandemic because public health measures may have reduced consumption from restaurants. As noted,

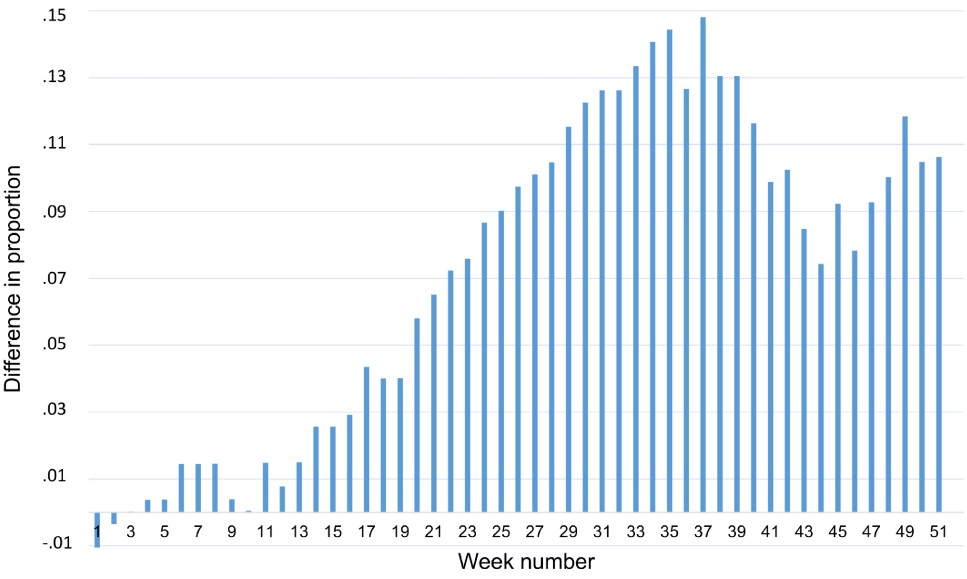

**Fig. 3 Difference in Regular Smokers' Quitting Behavior in 2020 vs. 2019.** Difference of cumulative proportion of quitting behavior for regular smokers in 2020 vs. 2019. Regular smokers in 2019 (2020) are defined as buying an average of at least one cigarette per day in the first ten weeks of 2019 (2020). $N = 325$.

restaurants were closed for in-person dining during the strict lockdown and were under occupancy restrictions for the remainder of the year. Shifting consumption patterns are less of a concern for cigarette purchases than for other grocery items because lockdown and social distancing measures did not differentially affect where people could buy tobacco: both convenience shops and supermarkets remained open throughout the pandemic and cigarettes cannot be purchased at restaurants.

Our data do not include purchases at convenience stores and so there may be concern that the decline in cigarette purchases we measure partially reflects a shift towards greater purchasing at corner shops and less purchasing at grocery stores. We report national administrative data on aggregate monthly sales from 2019–2020 separately for convenience stores and for grocery stores (Supplementary Fig. S2, monthly sales are indexed to 2015 sales). We find that during COVID-19 (compared to the same period in 2019), aggregate sales at convenience stores increased by an estimated 1.5 percent while sales at grocery stores increased by 5.5 percent. These results suggest that if anything there was a shift in purchases away from corner shops and towards supermarkets during the pandemic.

Relatedly, our data largely include in person sales at grocery stores and so our results could also reflect a shift towards greater purchasing online and less in person purchasing of cigarettes. Our data do include an online supermarket, which allows us to examine the share of cigarettes purchased at the online outlet over time. The share of cigarettes purchased at the online supermarket in our data, 3.2%, is similar to the national share of online supermarket purchases, 3.6% of all grocery purchases[32].

In our data, we find that the share of cigarette purchases made at the online supermarket declines by 27% during the COVID-19 pandemic compared to the same period in 2019 (from 3.20% of cigarette purchases to 2.34%).

An additional limitation of our dataset is that it does not include cash purchases. Although stores in Denmark by law have to accept cash payments, including during the pandemic, we note that cash purchases have been steeply declining in Denmark, from 48% of purchases in 2009 to only 16% of transactions in 2019[33]. If the shift away from cash continued through 2020 our estimates would be upwardly biased.

Finally, our findings align with aggregated national sales data, which show an 18% decline in cigarette sales in 2020 compared to 2019 and a 2.1% increase in alcohol sales (Supplementary Fig. S3). We therefore interpret our results as reflecting declines in overall cigarette purchases during the COVID-19 pandemic.

**Mechanisms**. We consider several potential mechanisms for the decline in tobacco purchases during COVID-19. It could be due to a range of factors, including but not limited to, the increased risks from COVID-19 that smokers face, changes in social and work environments, or financial concerns.

Our findings do not provide strong evidence that the declines are driven by changes to social and work environments. We find smaller declines in purchases during the strict lockdown when there was the largest change to social and work environments, and the largest declines during the spring and summer when the economy reopened. We also find increases in purchase quantities among occasional smokers, who are likely to be those most sensitive to "social smoking." These findings suggest social distancing measures and related restrictions are not driving the overall decline in smoking.

We find evidence that financial concerns may be partially driving the decline in tobacco use but do not fully explain it. Comparing estimates with and without controlling for price, we estimate that prices explain about 25% of the total decline in cigarette use during the pandemic. Using our data, we find evidence of price increases starting in week 27 of 2020 through the end of the year (see Supplementary Fig. S4). The increase is due to a national tax increase on cigarettes which came into effect on April 1st (week 14) but did not immediately affect prices due to retailers stocking up tobacco packages subject to the lower taxation rate[34]. We estimate that prices increased by about 29% percent compared to the same period in 2019, and a price elasticity of 0.55 among regular smokers – which is in line with existing literature[35]. We conduct a sensitivity analysis examining the effects of the pandemic prior to the price change in week 27 and find a similar impact of COVID-19 on cigarette purchases (Supplementary Table S7).

To examine the net effect of price and quantity changes, we estimate the impact of the pandemic on money spent on cigarettes (Supplementary Table S8). We find significant reductions in both the full population and all subgroups of smokers, with regular smokers spending an estimated $3.15 less per week on cigarettes ($p < 0.001$), a 24% decrease compared to an average of $12.87 during the same period in 2019. We conclude that price is partially driving the decrease in purchases but does not fully explain the decline. Relatedly, changes in financial security could affect demand for cigarettes. We are not able to control for changes in income (or other time-varying characteristics), but we note that in Denmark the overall unemployment rate only increased from 3.7% in 2019 to 4.6% in 2020 and salaries grew 2.5% (from the official statistics from Statistics Denmark).

Finally, several studies report growing awareness and fear of the increased risks of COVID-19 among smokers[36,37]. Beginning in March 2020, local media repeatedly highlighted the correlation between smoking and more severe COVID-19 infections. These reports may have encouraged tobacco cessation for consumers[38], though it is not clear how quickly and clearly information on increased risks of smoking spread to the public. For example, early in the pandemic there was media coverage suggesting that smoking may help protect against COVID-19[39] as well as the reverse[40]. We also find evidence, based on Google searches, of increased interest in the relationship between smoking and COVID-19 during this period. See Supplementary Table S9 and Figure S5 for a summary of the most relevant media publications in Denmark and a time trend of Google searches of related terms, respectively. Our data do not include measures of risk perceptions related to smoking and COVID-19, and so we cannot directly test for this mechanism in our population. However, two studies find evidence in the U.S. that smokers were on average more motivated to quit in response to the health risks of COVID-19[38,41].

Many other unobserved mechanisms might explain the declining cigarette purchases, including changes in social norms or growing concerns about exposing others to passive smoking. To the extent that our findings differ from prior work, it could be due to differences in methodology as well as context. Of particular relevance in the Danish context may be the high levels of trust and confidence in the national government, which could increase responsiveness to new information about increased health risks from smoking. In this regard, a 2022 OECD study finds that the share of people who report having confidence in the national government in Denmark is 71.6%, compared to 46.5% in the U.S.[42].

## Discussion
We document sustained declines in cigarette purchases throughout the pandemic. Our estimated purchase rate declines of about 25% in the full population and about 30% among regular smokers are large relative to policy interventions to reduce smoking. A recent review of these interventions -- including

informational campaigns, price increases, and smoking bans -- estimates a 7–8% reduction in cigarette demand, compared to our estimates of about 12%; and 3–4 percentage point increases in smoking cessation compared to about a 10-percentage point increase in quitting behavior during the pandemic[43]. One potential reason for the large impact of COVID-19 relative to other policy interventions is that – similar to smoking during pregnancy – COVID-19 makes the health risks of smoking more immediate. This may help some smokers overcome a key barrier to quitting – that the enjoyment of smoking is felt in the present and health costs are usually felt in the future.

Whether smoking returns to pre-pandemic levels is an open question. Studies of health shocks, such as pregnancy, demonstrate large rebound effects after the temporary shock fades away: e.g., 23% had ceased smoking during pregnancy but only 8% had quit smoking after pregnancy in Haug et al.[11]. At the same time, the pandemic has shifted some lifestyle habits in ways that people expect to persist[44]. If the health risk perception related to COVID-19 and smoking lasts long enough, the quitting behavior might also persist, loosening the addictive habit. Recent work estimates that decreasing longer-term smoking rates by 8 percent leads to a 6 percent reduction in mortality[45]. If the decline in smoking we document persists, not only could it help decrease the risks from COVID-19 as new variants emerge, but also have meaningful, longer-term benefits on population health and life expectancy beyond the pandemic.

**Reporting summary**. Further information on research design is available in the Nature Research Reporting Summary linked to this article.

## Data availability
The data that support the findings of this study are not publicly available as dictated by contractual agreement between the University of Copenhagen and the data collecting partner company. Data are however available from the authors upon reasonable request and with permission of the partner company, but restrictions apply to the availability of these data. The raw data associated with the figures is not publicly available as dictated by contractual agreement between the University of Copenhagen and the data collecting partner company Such data was made available to peer—reviewers.

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

## Author contributions

Conceptualization (T.R.F., A.P., S.S.); Methodology (T.R.F., A.P., S.S.); Investigation (T.R.F., A.P., S.S.); Visualization (T.R.F., A.P., S.S.); Data managing (T.R.F., A.P.); Writing – original draft (T.R.F., A.P., S.S.); Writing – review & editing (T.R.F., A.P., S.S.).

## Competing interest

The authors declare no competing interests.
