## [Peer Review File · Communications Medicine]

Reviewers' comments:

Reviewer #1 (Remarks to the Author):

This paper tackles an important question: whether the COVID 19 pandemic has resulted in a decrease in cigarette consumption.

There are several things I really like about this paper. First, the paper uses a unique dataset that allows the authors to study how purchases evolved through 2019 and 2020. Second, using the pre-pandemic data they can differentiate their results based on how much (and whether) people smoked at baseline. The paper is well-written and the empirical analysis is well-executed. I do not think there is any comparable study. Other papers investigating similar questions do not have access to such a rich dataset. I therefore think the paper should be published.

I don't have any substantial comments for the authors. I only have one small comment:

It would be useful if you could explain in more detail in the paper that your results rest on the assumption that there was no differential purchasing from grocery stores vs. corner shops before vs. during the pandemic. As you mention in your paper, they both remained open during the pandemic, so I find it convincing that we shouldn't be expecting a shift, but I think it would be good if you could mention that explicitly given that you only have data for the grocery store purchases. I am supposing that online purchasing of tobacco (from other retailers) is not possible in Denmark. It would be good if you could please clarify that.

Reviewer #2 (Remarks to the Author):

Comments on "Sustained Decline in Cigarette Consumption During COVID-19"

General Comment:

The study does an excellent job presenting evidence on how the COVID-19 pandemic might have changed cigarette and alcohol purchases patterns/behavior of individuals in Denmark. The following are some comments to improve the study.

Specific Comments:

1. In what units are purchase rate and quantity measured? Just the frequency? Packs? Cartons? Sticks? Do you observe total expenditure on cigarettes? If available, focusing on inflationary-adjusted expenditure and share (in Table S5) may be more interesting than the average purchase rate and quantity.
2. One threat to the validity of the results is the 8% prevalence rate constructed from individual-level purchases data, while national data shows about 17%. Consider reconstructing regular smokers to purchases per week or month as many people do not make trips to stores every day but purchase them in bulk periodically.
3. How is the regression specification on page 15 is a fixed effect model? I don't see the week and individual fixed effect dummies. Or did you demean the variables?
4. Model specification: The data as described is a time series consisting of weekly purchases from 2019 to 2020. However, I do not see any linear time trends to capture the underlying trends in weekly purchases. My understanding is that you tried controlling for these trends using the year

indicator (σ) since the trends in purchases in Weeks 1 -11 were similar between 2019 and 2020, but I am not sure how it would capture the underlying trends, provided the trends in 2019 is both nonlinear and decreased from March to December. Given the nonlinearity in the underlying trends of purchases (see 2019 trends in Figure 1), consider modeling it. In alternative specifications, you can include linear time trends (or even quadratic form) interrupted at the onset of the COVID-19 pandemic (week 12 in 2020 only) and week 20 in both 2019 and 2020. That is, include a linear time trend and interact with the COVID-19 dummy ($\gamma\sigma$ = March to December 2020) and another dummy for weeks 20 to 52 in both years.

5. In Model Specification 1, how should we interpret the coefficient β_2 ? The way the data is structured, with results presented in Tables S2-S7, is confusing. For example, COVID-19 is defined to include March to December 2019, whereas there was no COVID-19 during this period. The only variable that captures the COVID-19 period is $\gamma\sigma$.

6. Did the model control for individual-level factors? Does the individual-level data have unique identification numbers, allowing you to track their purchasing behavior over the two years? If yes, can you include individual fixed effects in the model?

7. Why is a difference-in-difference model discussed on page 15? Even if you want to use it, who would be in the treatment and control groups? Are you not presenting estimates from interrupted time series since everyone was treated in 2020 and there is no control group?

8. Contribution: While this present study is important and contributes to the literature, little is done to demonstrate it. If able to control for individual fixed effects, I suggest the authors discuss the importance of controlling for individual time-invariant characteristics in their model and compare how the national and state-level estimates may not capture individual-level effects. How about time-varying factors, such as income, education, ...? If not controlled, can you mention them as limitations?

9. How are cash and online purchases included in the data? Consider providing some limitations of the study to include no online and cash purchases, no control group, geographic locations, or individual characteristics, etc., if not accounted for in the model. Did the model account for these important factors?

10. On the readability, the study needs extensive revisions to correct several grammatical errors and incomplete sentences. For example, the sentence on page 5, paragraph 3, line 4: "...with the exception of..." is incomplete. Another example is that findings in references 21 and 22 were incorrectly cited. Vertical dotted lines were referred to in Figure 1, but I could not find them. Please move the Materials and Methods section to the section before the Results unless it is the formatting style of the journal.

Reviewer #3 (Remarks to the Author):

This paper uses novel data from an app that allows individuals to track their grocery purchases to understand patterns of cigarette purchases and alcohol purchases in Denmark during 2020, much of which was marked by the COVID-19 pandemic, relative to 2019. In contrast to much of the emerging literature, which finds increases in tobacco and alcohol purchases and consumption during the pandemic, the paper finds marked reductions in tobacco purchases, particularly among regular smokers and no change in alcohol purchases. Those studies are not based in Denmark and so it is entirely possible the experience is just different in Denmark. That said, one is left wondering if a major part of the discrepancy may be due to the specific data analyzed. That is, as I understand the data, they do not capture purchases made outside of grocery stores. If the pandemic changed where

individuals make their purchases generally or where they purchased specific items, then these data may provide a misleading picture of tobacco purchases. For example, if individuals are less likely to go to the grocery store and/or more likely to make cigarette purchases at convenience stores then actual purchases may have changed in ways not captured in the data. While the paper tries to dismiss this concern by arguing that lockdowns and social distancing did not differentially affect where people could buy tobacco or alcohol, it may have changed where people opted to make these purchases. Frequent purchases of cigarettes at smaller convenience stores, for example, could have crowded out purchases at grocery stores. Without other data – e.g., survey measures, sales data, etc. - to validate the findings, I worry that the data from this app provide a misleading picture of what is happening to tobacco and alcohol consumption in Denmark.

Other Comments:

1) The authors interpret the patterns they observe in the data as reflecting concerns about health risk, specifically concerns that tobacco consumption puts one at higher risk for adverse COVID-19 outcomes. However, the data really do not provide any measure of risk perceptions and many other factors put individuals at higher risk of adverse outcomes, including obesity. Yet, other data suggests a decline in exercise. While the paper's analysis of mechanisms provides some support for this interpretation, it really seems overly strong to attribute these patterns to concerns about the risks of smoking.

2) The paper adds cigarette prices to some of the models and concludes that some of the changes in consumption are due to price. If the pandemic really caused a shock to demand, however, shouldn't prices have declined? Were there supply shocks due to the pandemic?

3) The paper says that the app data capture spending on grocery purchases as well the environmental impact of the purchases. What does the environmental impact of the purchases mean?

4) In the notes to the tables, it would make sense to clarify that the COVID-19 indicator captures weeks 11 on in any given calendar year.

5) There are many other papers on the topic now, including ones based in Europe, that should probably be added to the citations. Some examples include:

<https://academic.oup.com/eurpub/article/31/5/1076/6214519>

<https://www.sciencedirect.com/science/article/pii/S2352340921007617>

<https://www.mdpi.com/1660-4601/18/13/7128>

<https://www.ncbi.nlm.nih.gov/pmc/articles/PMC7643580/>

<https://www.ncbi.nlm.nih.gov/pmc/articles/PMC7386200/>

Note that other work seems to align more with the findings from Denmark:

<https://www.sciencedirect.com/science/article/pii/S0306460321001027>

<https://academic.oup.com/ntr/article/22/9/1662/5826329>

Reviewer #4 (Remarks to the Author):

Review of COMMSMED-21-0589

The current paper examines individual level cigarette purchase data among a sample of the Danish population before versus during the COVID-19 pandemic. The authors find a decrease in cigarette purchase rates among their sample during the pandemic. They further demonstrate that this decline is largely driven by “regular smokers” and that it cannot be fully attributed to changes in price. The writing is clear and the research question, and data, is important and interesting. I do have some concerns with the presentation of measures, the discussion of potential mechanisms, and discussion/conclusion which I outline below.

o I have several comments on the measures/definitions used:

- The authors claim that their data provides evidence for declining consumption based on purchase rates. While purchase rates may serve as a proxy for consumption, this needs to be made clearer (vs. a blanket claim that consumption declined). I recommend the authors either directly say at the outset that purchase rates serve as a proxy for consumption or simply discuss purchase rates throughout the paper (and save their proposition about decreased consumption for the discussion/conclusion). Citations where purchase rates have been used as a measure of consumption would be helpful. Further, the authors should add a few sentences (perhaps in a footnote or discussion) explaining that their data does not allow them to use more common definitions of current smokers (e.g., smoked at least 100 cigarettes in lifetime, etc.).

- In the main analyses smokers are defined as anyone who buys at least one cigarette per day. The authors should either cite other research that has used this same measure, or provide discussion about how this differs from typical classifications of smokers and why (a more conservative approach would be to refer to these individuals as “people who purchased cigarettes” in the main analysis and then save smoker classification for the occasional/regular smokers in the secondary analyses).

o The introduction starts out by discussing the health costs of smoking, and how these health costs have increased during the pandemic. There are two elements of this that could be improved (the latter being a larger change than the former): (1) “health costs of smoking” should be defined at its first mention on page 4—the way it is discussed feels more like health risks than health costs (which would also imply financial costs); (2) the introduction of the paper is framed such that the authors propose that increased awareness of health costs/risks is a main driver for the decrease in purchase rates. The discussion of potential mechanisms at the end of the paper, however, focuses on the increased cost of cigarettes (which I agree should be thoroughly discussed) and lists health risks last with very little discussion. This makes the front-end and back-end of the papers feel mismatched. The front-end led me to believe that health risks would not only be a potential explanation for the authors’ findings, but that the paper would provide a thorough discussion of why that may be. If the authors want to keep the framing of the paper as one about awareness of health risks, then the discussion/evidence as to why this is a relevant explanation could be strengthened.

o Related to the above, on pg. 4 the authors mention prior research that shows that health shocks (e.g., pregnancy) decrease smoking. This is a key point and I would encourage the authors to consider highlighting this more.

o It would be helpful to have more information on the sample used. For instance, how do people know/sign-up for the app? Are they compensated? Is there population level data on the average % of cigarettes purchased at these locations vs. other locations that are not in the data?

o On pg. 5 the authors briefly discuss surveys on consumption changes during the pandemic and

indicate that most of this data finds that people report an increase in tobacco consumption. I have several recommendations for this section. First, the authors cite four surveys in a way that implies a comprehensive list. There have been many more surveys/studies conducted on tobacco consumption during the pandemic and the text should reflect this. Second, the location/population of the surveys cited should be mentioned. Finally, as the authors suggest, many of these surveys find the opposite of what the authors find: an increase in tobacco consumption. However, the authors do not discuss why they believe their data shows a different pattern—is it that they use individual level data and these others use population level data (and if so, why would that lead to this difference?). Do they believe that there was a specific campaign that occurred in Denmark that made people more aware of the health risks than in other countries (as health risks is their main argument as to why this decline occurred), or might it have to do with price changes in Denmark that may not have occurred in other countries?

- o The authors may consider adding a sentence or footnote on page 8 indicating that they will return to the discussion of price in the discussion section (as the results from the analyses raise questions).

- o It would be nice to see the sample sizes for the different smoker groups mentioned alongside the analyses on page 8 (rather than just in the appendix).

- o The authors should add citations to support their claim that alcohol could be a substitute for cigarettes as there is much evidence suggesting alcohol is a complement, not a substitute, for tobacco.

- o The occasional smokers show an increase in purchase quantity—this should be discussed. Further, given this, the claim on page 9 that regular smokers account “for the full decrease in cigarette consumption” is inaccurate.

- o The authors should acknowledge that they discuss some, but not all, potential mechanisms (e.g., on page 10 the sentence should say something along the lines of “it could be due to a range of factors, including but not limited to...”

- o In the discussion of price increases (pg. 11), the authors may also want to mention simultaneous decreases in wages/unemployment as a result of the pandemic.

- o The discussion on mechanisms falls a bit flat. First, as mentioned above, it feels mismatched with the intro/framing of the paper being about health risks. Second, there should be some mention that these are just a few possibilities and that there may be other potential mechanisms at play (e.g., shift in social norms during the pandemic, fear not only for one’s own health but for others, etc.). Finally, this would also be a good place to discuss mechanisms that are specific to Denmark and could explain the discrepancy in findings between this data and the surveys that find an increase in consumption in other countries.

- o The authors might consider adding a table to the appendix that provides more detail on the review of interventions mentioned at the bottom of pg. 11.

- o Pg. 12. The last sentence in the second paragraph could be broadened – not only would a continued decline protect against risks from new variants, but it could have a meaningful effect on population health and life expectancy beyond covid.

- o The authors compare the smoking rate in their sample to that of the Danish population. If possible, it would be nice to see a comparison of the smoking rate in this sample compared to a matched demographic sample (as the sample used in this study differed from the general population in ways that may affect smoking rates, such as gender and income).

- o Another possibility that the authors may want to mention is whether people may have started purchasing cigarettes online during this time (and whether or not that is prohibited by law).

- o Much of the tobacco literature on purchase rates uses population level data. The authors may wish to highlight the uniqueness of their individual-level data.

Minor Comments:

- o The background section in the abstract should have “use” added to the end of the last sentence.
- o There should be a citation added to the first sentence after “leading causes of death globally...”
- o The following sentence is not clear: “Tobacco products are highly addictive with extremely low rates of cessation, particularly for sustained periods.”
- o In several places, the authors end the sentence with a citation rather than listing out the name of the cited survey (e.g., on pg. 5 in the last paragraph they end a sentence with “with the exception of 20”). It would be more fluent to also add the name of the survey to the sentence.

I wish the authors the best of luck as they move this important research forward.

Reviewers' comments:

Reviewer #1 (Remarks to the Author):

This paper tackles an important question: whether the COVID 19 pandemic has resulted in a decrease in cigarette consumption.

There are several things I really like about this paper. First, the paper uses a unique dataset that allows the authors to study how purchases evolved through 2019 and 2020. Second, using the pre-pandemic data they can differentiate their results based on how much (and whether) people smoked at baseline. The paper is well-written and the empirical analysis is well-executed. I do not think there is any comparable study. Other papers investigating similar questions do not have access to such a rich dataset. I therefore think the paper should be published.

Thank you for your comments about the contribution of our paper.

I don't have any substantial comments for the authors. I only have one small comment:

It would be useful if you could explain in more detail in the paper that your results rest on the assumption that there was no differential purchasing from grocery stores vs. corner shops before vs. during the pandemic. As you mention in your paper, they both remained open during the pandemic, so I find it convincing that we shouldn't be expecting a shift, but I think it would be good if you could mention that explicitly given that you only have data for the grocery store purchases. I am supposing that online purchasing of tobacco (from other retailers) is not possible in Denmark. It would be good if you could please clarify that.

This is an important point which we now give more emphasis. We have added a new section that discusses concerns about potential shifts in purchasing behavior. To address the concern about purchases shifting to corner shops during the pandemic, we now include national administrative data showing aggregate sales by month at convenience stores. To address concerns about a shift to online purchases, we examine the share of cigarettes purchased at an online store in our data. We also report aggregated national sales data. We have pasted the relevant section of the text and new figures below:

Our data do not include purchases at convenience stores and so there may be concern that the decline in cigarette purchases we measure partially reflects a shift towards greater purchasing at corner shops and less purchasing at grocery stores. We report national administrative data on aggregate monthly sales from 2019-2020 separately for convenience stores and for grocery stores (Appendix Figure S3, monthly sales are indexed to 2015 sales). We find that during COVID-19 (compared to the same period in 2019), aggregate sales at convenience stores increased by an estimated 1.5 percent while sales at grocery stores increased by 5.5 percent. These results suggest that if anything there was a shift in consumption away from corner shops and towards supermarkets during the pandemic.

Relatedly, our data largely include in person sales at grocery stores and so our results could also reflect a shift towards greater purchasing online and less in person purchasing of

cigarettes. Our data do include an online supermarket, which allows us to examine the share of cigarettes purchased at the online outlet over time. The share of cigarettes purchased at the online supermarket in our data, 3.2%, is similar to the national share of online supermarket purchases, 3.6% of call grocery purchases.

In our data, we find that the share of cigarette purchases made at the online supermarket declines by 27% during the COVID-19 pandemic compared to the same period in 2019 (from 3.20% of cigarette purchases to 2.34%).

An additional limitation of our dataset is that it does not include cash purchases. However, we note that cash purchases have been steeply declining in Denmark, from 48% of purchases in 2009 to only 16% of transactions in 2019³². If the shift away from cash continued through 2020 our estimates would be upwardly biased.

Finally, our findings align with aggregated national sales data, which show an 18% decline in cigarette sales in 2020 compared to 2019 and a 2.1% increase in alcohol sales (Appendix Figure S4). We therefore interpret our results as reflecting declines in overall cigarette purchases during the COVID-19 pandemic.

Figure S3. Convenience store and grocery store sales 2019-2020 (Index: 2015=100)

Note: The total amount of monthly sales in convenience stores and grocery stores (in Danish: “Købmand og døgnskiosker” & “Detailomsætningsindex”), indexed to average monthly sales in 2015. Data source: Statistics Denmark, www.statistikbanken.dk/DETA151 + www.statistikbanken.dk/DETA152.

Figure S4. National cigarette and alcohol sales by year in Denmark

Note: The panel to the left shows the number of cigarettes that an average Danish adult bought in 2019 and 2020. The panel on the right shows the liter of pure alcohol an average Danish adult bought in 2019 and 2020. The pure alcohol measure is determined by calculating the alcohol share (in liters) of each alcohol-containing product sold. Data source: Statistics Denmark: www.statistikbanken.dk/ALKO2

Reviewer #2 (Remarks to the Author):

Comments on “Sustained Decline in Cigarette Consumption During COVID-19”

General Comment:

The study does an excellent job presenting evidence on how the COVID-19 pandemic might have changed cigarette and alcohol purchases patterns/behavior of individuals in Denmark. The following are some comments to improve the study.

Thank you for your support for the study and for your helpful comments.

Specific Comments:

1. In what units are purchase rate and quantity measured? Just the frequency? Packs? Cartons? Sticks? Do you observe total expenditure on cigarettes? If available, focusing on inflationary-adjusted expenditure and share (in Table S5) may be more interesting than the average purchase rate and quantity.

We collect cigarette data measured as packages of 20 cigarettes (the only available format in Denmark). In our analysis, for quantity, we decided to look at one cigarette as the unit, given that in some countries the packages vary in how many cigarettes they contain and given the definition of regular smokers found in the literature and applied in our categorization. We now include the following in the Materials and Methods section to clarify our measures and it reads:

Our data include the date of each purchase and the number of cigarette packages purchased if any – in Denmark, cigarettes are only available in packages of 20 cigarettes. We aggregate all purchases made in a given week and analyze weekly cigarette purchase rates and quantities. Weekly cigarette purchase rates are the share of individuals who purchased at

least one package of cigarettes in a given week. We measure weekly cigarette quantity as the total number of cigarettes purchased in a week, which is equivalent to 20 times the number of packages purchased.

The inflation in Denmark during the period of the study was limited (0.8 in 2019 and 0.4 in 2020, according to Statistics Denmark www.dst.dk), and we have therefore decided not to adjust expenditures for inflation. Furthermore, our analysis controls for the nominal price.

2. One threat to the validity of the results is the 8% prevalence rate constructed from individual-level purchases data, while national data shows about 17%. Consider reconstructing regular smokers to purchases per week or month as many people do not make trips to stores every day but purchase them in bulk periodically.

Our measure is not affected by bulk purchases because we classify regular smokers based on their yearly consumption. We now clarify this in the Materials and Methods section as follows:

Among smokers ($n=983$), we also characterize regular smokers as those who purchased the equivalent of at least one cigarette per day in 2019 ($n=325$, 33% of smokers, 8% of our sample) and occasional smokers who purchased at least one cigarette in 2019 but less than the equivalent of one cigarette per day ($n=658$, 66.9% of smokers, 16.3% of our sample).¹

Footnote 1: According to the National Institute for Health Education Risk Prevention (INPES), a regular smoker is somebody who admits to smoking at least one cigarette (or equivalent) per day. To calculate average daily cigarette purchases, we divide an individual's total cigarette purchases in 2019 by 365. Our data do not allow us to use another common survey measure of current smoking: having ever smoked 100 or more cigarettes within one's lifetime and smoking every day or some days at the time of survey.

Our data come from a sample that is not nationally representative, which may be the reason for the lower share of regular smokers in our sample compared to national averages. We now include analyses that uses inverse probability weighting (IPW) to re-weight our sample based on the share of smokers in the population. We report the results in Appendix Table S8 and have pasted the table below. We now include the following discussion in the Materials and Methods section (without citations):

The proportion of regular smokers in our data is lower than in the national Danish population: an estimated 17% of Danes smoke daily 28. To address the non-representativeness of our sample, we include a re-weighted analysis that uses inverse probability weighting to match the shares of smokers and non-smokers observed in the national population, and the results do not change (Appendix Table S8).

Appendix Table S8. Regression on the impact of COVID-19 on cigarette quantity using Inverse Probability Weighting of the sample for the national shares of smokers and non-smokers.

IPW on cigarette quantity by smoker category										
	All	All+price	Non-smokers	Non-smokers+price	Smokers	Smokers+price	Occasionals	Occasionals+price	Regulars	Regulars+price
Spring	0.397*** (0.134)	0.193 (0.138)	0.002 (0.032)	-0.006 (0.027)	1.624*** (0.543)	0.812 (0.560)	0.333 (0.205)	0.225 (0.208)	4.237*** (1.579)	2.001 (1.640)
Summer	-0.012 (0.135)	0.084 (0.131)	-0.034 (0.027)	-0.030 (0.027)	0.055 (0.547)	0.439 (0.533)	0.077 (0.236)	0.128 (0.228)	0.012 (1.587)	1.068 (1.544)
Fall	-0.239* (0.132)	-0.050 (0.129)	-0.029 (0.026)	-0.021 (0.027)	-0.892* (0.537)	-0.142 (0.524)	0.010 (0.240)	0.110 (0.220)	-2.720* (1.547)	-0.652 (1.522)
Week 11 - 52	0.630*** (0.168)	0.587*** (0.165)	0.018 (0.024)	0.017 (0.025)	2.535*** (0.684)	2.362*** (0.669)	-0.114 (0.256)	-0.137 (0.253)	7.899*** (1.971)	7.423*** (1.930)
Year (2020)	0.215 (0.175)	0.221 (0.176)	0.097*** (0.019)	0.098*** (0.019)	0.582 (0.719)	0.605 (0.721)	0.176 (0.253)	0.179 (0.253)	1.403 (2.114)	1.467 (2.122)
COVID-19 period ^	-0.949*** (0.215)	-0.400** (0.188)	0.059** (0.025)	0.081** (0.037)	-4.083*** (0.875)	-1.897** (0.764)	0.542* (0.290)	0.834*** (0.297)	-13.447*** (2.503)	-7.424*** (2.203)
Cigarettes price		-3.213*** (0.679)		-0.131 (0.138)		-12.802*** (2.736)		-1.710 (1.079)		-35.260*** (7.845)
Constant	2.784*** (0.221)	14.686*** (2.532)	0.000 (.)	0.486 (0.512)	11.449*** (0.851)	58.874*** (10.164)	1.663*** (0.143)	7.998** (3.990)	31.262*** (2.180)	161.879*** (28.859)
N	420368	420368	318136	318136	102232	102232	68432	68432	33800	33800
Clusters (individuals)	4042	4042	3059	3059	983	983	628	628	325	325

* p<0.1, ** p<0.05, *** p<0.01

^ The variable COVID-19 period represents the interaction Week 11 - 52 X Year (2020)

3. How is the regression specification on page 15 is a fixed effect model? I don't see the week and individual fixed effect dummies. Or did you demean the variables?

Thank you for careful read of the regression specification. We estimate individual fixed effects with error terms clustered at the individual level and include seasonal fixed effects (rather than week fixed effects). We have revised the regression specification to include the individual fixed effects. We have pasted below the regression specification and the description related to the fixed effects:

We estimate changes in purchases during COVID-19 using the following regression with individual fixed effects:

$$y_{it} = \alpha_i + \beta_1 \sigma + \beta_2 \gamma + \beta_3 \gamma \sigma + \beta_4 \log(\varpi_t) + \beta_5 x_1 + \beta_6 x_2 + \beta_7 x_3 + \varepsilon_{it}$$

where the dependent variable y_{it} is the outcome of interest for individual i at the weekly level t ; the dummy σ is an indicator for the year 2020 (vs. 2019); the dummy γ is the time indicator for weeks 11 - 52; three covariates x_k control for seasonal fixed effects – where $k=1$ corresponds to spring (week 12 – week 25), $k=2$ to summer (week 26 – week 38) and $k=3$ to fall (week 39 – week 52). The individual fixed effect is represented by α_i . Our coefficient of interest, β_3 , estimates the term interacting the indicator for 2020 with the indicator for the weeks 11 – 52, which is the COVID-19 period (in the tables we label the coefficient “COVID-19 period”). We include the logarithmic function of ϖ_t to control for average weekly price as measured in our data ³⁷; finally, ε_{it} is the residual term.

4. Model specification: The data as described is a time series consisting of weekly purchases

from 2019 to 2020. However, I do not see any linear time trends to capture the underlying trends in weekly purchases. My understanding is that you tried controlling for these trends using the year indicator (σ) since the trends in purchases in Weeks 1 -11 were similar between 2019 and 2020, but I am not sure how it would capture the underlying trends, provided the trends in 2019 is both nonlinear and decreased from March to December. Given the nonlinearity in the underlying trends of purchases (see 2019 trends in Figure 1), consider modeling it. In alternative specifications, you can include linear time trends (or even quadratic form) interrupted at the onset of the COVID-19 pandemic (week 12 in 2020 only) and week 20 in both 2019 and 2020. That is, include a linear time trend and interact with the COVID-19 dummy ($\gamma\sigma = \text{March to December 2020}$) and another dummy for weeks 20 to 52 in both years.

Given the seasonality in purchasing patterns, our main specification controls for time trends using seasonal fixed effects as discussed in response to comment above. As you note, our main specification does not control for aggregate trends across years. In the estimates below, we have followed your suggestions for specifications that include a linear time trend (Tables R.1) and a quadratic time trend (Tables R.2). We include a control for week number starting with week 1 of 2019 through week 105, the final week of 2020. In the specification with the quadratic form, we include a control for week number squared. Our estimates for the impact of COVID-19 do not change.

We have also included Figure R.1 below showing cigarette quantity plotted from week 1 to week 105. We separately plot linear trend lines for the pre-COVID-19 period (week 1 of 2019 through week 10 of 2020. Weeks 1-63) and the COVID-19 period (weeks 11-52 of 2020, weeks 64-105). As shown in the figure, the linear trend in the pre-COVID-19 period is slightly positive while the linear trend in the COVID-19 period is steeply negative. We do not currently include these sensitivity checks in the Appendix of the paper, but we are happy to add either or both of them if you and the editor prefer.

Tables R.1. Regression on the impact of COVID-19 on cigarette consumption with linear time trends.

Panel A – Purchase Rate

F.E. clustered regressions

	All	All	Non-smokers	Non-smokers	Smokers	Smokers	Occasionals	Occasionals	Regulars	Regulars
Week number	-0.000** (0.000)	-0.000 (0.000)	0.000 (0.000)	0.000 (0.000)	-0.001*** (0.000)	-0.000 (0.000)	-0.000 (0.000)	-0.000 (0.000)	-0.001** (0.000)	-0.000 (0.001)
Spring	0.001 (0.001)	0.000 (0.001)	-0.000 (0.000)	-0.000 (0.000)	0.003 (0.006)	0.002 (0.006)	0.005 (0.005)	0.005 (0.005)	-0.002 (0.014)	-0.005 (0.014)
Summer	-0.001 (0.001)	-0.000 (0.001)	-0.000 (0.000)	-0.000 (0.000)	-0.003 (0.005)	-0.001 (0.005)	0.007 (0.005)	0.007 (0.005)	-0.023* (0.014)	-0.017 (0.014)
Fall	-0.002 (0.001)	-0.002 (0.001)	-0.001 (0.001)	-0.001 (0.001)	-0.007 (0.006)	-0.006 (0.006)	0.003 (0.005)	0.004 (0.005)	-0.028** (0.014)	-0.026* (0.014)
Week 11 52	0.010*** (0.002)	0.008*** (0.002)	-0.000 (0.001)	-0.000 (0.000)	0.039*** (0.008)	0.032*** (0.008)	0.006 (0.007)	0.004 (0.007)	0.108*** (0.020)	0.087*** (0.020)
Year (2020)	0.009*** (0.003)	0.006** (0.003)	0.002*** (0.001)	0.002*** (0.001)	0.031*** (0.012)	0.018 (0.011)	0.014 (0.010)	0.012 (0.010)	0.066** (0.029)	0.029 (0.028)
COVID period	-0.015*** (0.002)	-0.013*** (0.002)	0.001 (0.000)	0.001 (0.001)	-0.065*** (0.009)	-0.054*** (0.008)	-0.008 (0.006)	-0.006 (0.006)	-0.183*** (0.023)	-0.151*** (0.020)
Cigarette price		-0.017*** (0.006)		-0.000 (0.002)		-0.070*** (0.025)		-0.012 (0.020)		-0.187*** (0.063)
Constant	0.045*** (0.001)	0.108*** (0.023)	-0.000 (0.000)	0.000 (0.007)	0.186*** (0.006)	1.442*** (0.090)	0.055*** (0.004)	0.098 (0.072)	0.452*** (0.016)	1.140*** (0.227)
N	420368	420368	318136	318136	102232	102232	68432	68432	33800	33800

d.v. any cigarette dummy
* p<0.1, ** p<0.05, *** p<0.01

Panel B – Quantity

F.E. clustered regressions

	All	All	Non-smokers	Non-smokers	Smokers	Smokers	Occasionals	Occasionals	Regulars	Regulars
Week number	-0.012** (0.005)	0.000 (0.005)	0.000 (0.001)	0.001 (0.001)	-0.049** (0.020)	-0.002 (0.022)	-0.009 (0.008)	-0.003 (0.008)	-0.132** (0.057)	0.000 (0.063)
Spring	0.240* (0.141)	0.193 (0.142)	0.003 (0.028)	0.000 (0.027)	0.981* (0.574)	0.792 (0.576)	0.219 (0.217)	0.194 (0.218)	2.524 (1.679)	2.003 (1.685)
Summer	-0.006 (0.134)	0.084 (0.131)	-0.034 (0.027)	-0.029 (0.027)	0.080 (0.547)	0.438 (0.532)	0.081 (0.236)	0.127 (0.228)	0.078 (1.585)	1.068 (1.544)
Fall	-0.076 (0.134)	-0.050 (0.133)	-0.029 (0.031)	-0.028 (0.031)	-0.225 (0.542)	-0.121 (0.539)	0.129 (0.247)	0.142 (0.244)	-0.941 (1.561)	-0.654 (1.554)
Week 11 52	0.943*** (0.209)	0.587*** (0.191)	0.018 (0.026)	0.002 (0.027)	3.821*** (0.851)	2.408*** (0.778)	0.115 (0.328)	-0.066 (0.310)	11.325*** (2.437)	7.417*** (2.246)
Year (2020)	0.852*** (0.294)	0.221 (0.272)	0.096* (0.053)	0.068 (0.045)	3.204*** (1.197)	0.698 (1.110)	0.644 (0.483)	0.323 (0.459)	8.387*** (3.472)	1.456 (3.228)
COVID period	-0.949*** (0.215)	-0.400** (0.188)	0.059*** (0.025)	0.084** (0.037)	-4.083*** (0.875)	-1.905** (0.765)	0.542* (0.290)	0.821*** (0.303)	-13.447*** (2.503)	-7.424*** (2.201)
Cigarette price		-3.213*** (0.727)		-0.147 (0.138)		-12.754*** (2.940)		-1.634 (1.133)		-35.265*** (8.463)
Constant	2.850*** (0.132)	14.686*** (2.636)	-0.000 (0.010)	0.542 (0.507)	11.721*** (0.540)	58.793*** (10.653)	1.711*** (0.198)	7.732* (4.161)	31.986*** (1.586)	161.908*** (30.631)
N	420368	420368	318136	318136	102232	102232	68432	68432	33800	33800

d.v. weekly number of cigarettes
* p<0.1, ** p<0.05, *** p<0.01

Tables R.2. Regression on the impact of COVID-19 on cigarette consumption with quadratic time trends.

Panel A – Purchase Rate

F.E. clustered regressions

	All	All	Non-smokers	Non-smokers	Smokers	Smokers	Occasionals	Occasionals	Regulars	Regulars
Week number	0.000 (0.000)	0.000 (0.000)	0.000 (0.000)	0.000 (0.000)	0.000 (0.000)	0.000 (0.000)	0.000 (0.000)	0.000 (0.000)	0.001 (0.001)	0.001 (0.001)
Week number^2	-0.000*** (0.000)	-0.000** (0.000)	-0.000 (0.000)	-0.000 (0.000)	-0.000*** (0.000)	-0.000** (0.000)	-0.000 (0.000)	-0.000 (0.000)	-0.000*** (0.000)	-0.000** (0.000)
Spring	-0.000 (0.001)	-0.000 (0.001)	-0.000 (0.000)	-0.000 (0.000)	0.000 (0.006)	0.000 (0.005)	0.004 (0.005)	0.008 (0.005)	-0.008 (0.015)	-0.008 (0.015)
Summer	-0.002 (0.001)	-0.002 (0.001)	-0.000 (0.000)	-0.000 (0.000)	-0.007 (0.006)	-0.006 (0.005)	-0.006 (0.005)	-0.005 (0.005)	-0.031** (0.014)	-0.028* (0.014)
Fall	-0.002* (0.001)	-0.002* (0.001)	-0.001 (0.001)	-0.001 (0.001)	-0.008 (0.006)	-0.008 (0.005)	-0.003 (0.005)	-0.003 (0.005)	-0.032** (0.014)	-0.031** (0.014)
Week 11 52	0.006*** (0.002)	0.006*** (0.002)	-0.000 (0.001)	-0.000 (0.001)	0.026*** (0.009)	0.026*** (0.008)	0.002 (0.008)	0.002 (0.008)	0.074*** (0.021)	0.074*** (0.021)
Year (2020)	0.004 (0.003)	0.004 (0.003)	0.002*** (0.001)	0.002*** (0.001)	0.009 (0.012)	0.009 (0.010)	0.009 (0.010)	0.009 (0.010)	0.009 (0.030)	0.009 (0.030)
COVID period	-0.010*** (0.002)	-0.010*** (0.002)	0.001 (0.001)	0.001 (0.001)	-0.042*** (0.009)	-0.042*** (0.007)	-0.002 (0.007)	-0.002 (0.007)	-0.122*** (0.022)	-0.124*** (0.022)
Cigarette price		0.006 (0.006)	-0.000 (0.002)	-0.000 (0.002)	-0.011 (0.023)	-0.011 (0.023)	0.012 (0.018)	0.012 (0.018)	-0.056 (0.058)	-0.056 (0.058)
Constant	0.044*** (0.002)	0.052** (0.022)	-0.000 (0.008)	-0.002 (0.008)	0.181*** (0.007)	0.222*** (0.085)	0.054*** (0.005)	0.012 (0.068)	0.439*** (0.018)	0.647*** (0.217)
N	420368	420368	318136	318136	102232	102232	68432	68432	33800	33800

d.v. any cigarette dummy
* p<0.1, ** p<0.05, *** p<0.01

Panel B – Quantity

F.E. clustered regressions

	All	All	Non-smokers	Non-smokers	Smokers	Smokers	Occasionals	Occasionals	Regulars	Regulars
Week number	-0.012** (0.005)	0.015 (0.010)	0.002 (0.001)	0.001 (0.001)	0.091** (0.040)	0.055 (0.039)	0.010 (0.016)	0.006 (0.017)	0.255** (0.117)	0.155 (0.112)
Spring	0.240* (0.141)	0.164 (0.145)	-0.002 (0.026)	-0.001 (0.026)	0.633 (0.589)	0.678 (0.591)	0.172 (0.218)	0.177 (0.217)	1.566 (1.726)	1.693 (1.732)
Summer	-0.006 (0.134)	-0.040 (0.145)	-0.040 (0.026)	-0.036 (0.025)	-0.425 (0.596)	0.028 (0.593)	0.013 (0.252)	0.063 (0.239)	-1.312 (1.729)	-0.044 (1.728)
Fall	-0.076 (0.134)	-0.091 (0.134)	-0.032 (0.030)	-0.030 (0.030)	-0.448 (0.551)	-0.281 (0.543)	0.099 (0.261)	0.117 (0.261)	-1.554 (1.581)	-1.088 (1.555)
Week 11 52	0.943*** (0.209)	0.460** (0.191)	-0.006 (0.030)	-0.006 (0.030)	1.800** (0.780)	1.911** (0.780)	-0.147 (0.334)	-0.143 (0.335)	5.982*** (2.247)	6.078*** (2.245)
Year (2020)	0.852*** (0.294)	0.027 (0.278)	0.057 (0.043)	0.056 (0.043)	-0.016 (1.133)	-0.064 (1.134)	0.210 (0.491)	0.204 (0.491)	-0.472 (3.282)	-0.606 (3.288)
COVID period	-0.949*** (0.215)	-0.143 (0.214)	-0.101** (0.051)	-0.099* (0.051)	-0.640 (0.870)	-0.897 (0.864)	1.006** (0.412)	0.978** (0.416)	-3.974 (2.490)	-4.691* (2.463)
Week number^2	-0.000 (0.000)	-0.000 (0.000)	-0.000 (0.000)	-0.000 (0.000)	-0.001** (0.000)	-0.001** (0.000)	-0.000 (0.000)	-0.000 (0.000)	-0.003** (0.001)	-0.002* (0.001)
Cigarette price		-1.956*** (0.612)		-0.073 (0.141)		-7.814*** (2.469)		-0.865 (1.161)		-21.882*** (7.031)
Constant	2.850*** (0.132)	9.956*** (2.252)	-0.000 (0.013)	0.264 (0.520)	10.995*** (0.634)	40.115*** (9.088)	1.613*** (0.223)	4.839 (4.343)	29.980*** (1.861)	111.537*** (25.849)
N	420368	420368	318136	318136	102232	102232	68432	68432	33800	33800

d.v. weekly number of cigarettes
* p<0.1, ** p<0.05, *** p<0.01

Figure R.1. Weekly cigarettes consumption over week with separate trend lines.

The graph illustrates the weekly cigarettes consumption over 2019 and 2020. The graph includes a separate trend line of the weekly consumption before and after the lockdown in week 11 of 2020. The slope of the trend line is 0.00186 ($p=0.510$) up until week 63, while is 0.38912 ($p=0.000$) after week 63.

5. In Model Specification 1, how should we interpret the coefficient β_2 ? The way the data is structured, with results presented in Tables S2-S7, is confusing. For example, COVID-19 is defined to include March to December 2019, whereas there was no COVID-19 during this period. The only variable that captures the COVID-19 period is $\gamma\sigma$.

We agree that the labels for the coefficients in Tables S2-S7 were confusing. We now make clear that our coefficient of interest is β_3 , which as you note captures the COVID-19 period by interacting $\gamma\sigma$. We have changed the coefficient labels in our tables to clarify the different variables. We now label β_2 “weeks 11-52”, which includes both 2019 and 2020. We label β_3 “COVID-19 period” and include in the notes that “COVID-19 period is the coefficient for the interaction of Week 11-52 x Year (2020).” We have added the following to the discussion of the regression specification:

Our coefficient of interest, β_3 , estimates the term interacting the indicator for 2020 with the indicator for the weeks 11 – 52, which is the COVID-19 period (in the tables we label the coefficient “COVID-19 period”).

6. Did the model control for individual-level factors? Does the individual-level data have unique identification numbers, allowing you to track their purchasing behavior over the two years? If yes, can you include individual fixed effects in the model?

Thank you for this comment, which was also brought up by another reviewer as well. As we discussed above in response to comment 3, we have now clarified that we include individual fixed effects in the model. We have also added emphasis on this feature of our data which other reviewers also highlighted as a contribution of our study.

7. Why is a difference-in-difference model discussed on page 15? Even if you want to use it, who would be in the treatment and control groups? Are you not presenting estimates from interrupted time series since everyone was treated in 2020 and there is no control group?

We have removed mention of the model as a difference-in-difference.

8. Contribution: While this present study is important and contributes to the literature, little is done to demonstrate it. If able to control for individual fixed effects, I suggest the authors discuss the importance of controlling for individual time-invariant characteristics in their model and compare how the national and state-level estimates may not capture individual-level effects. How about time-varying factors, such as income, education, ...? If not controlled, can you mention them as limitations?

Thank you for your guidance on the contribution of the paper. We have drawn on your language to add the following new paragraph to the end of the introduction:

Our study is the first to use individual-level data on tobacco consumption prior to and during the pandemic to examine responses to the COVID-19 pandemic and the associated restrictions. This allows us to control for individual time-invariant characteristics and to examine heterogeneous individual-level effects that may be masked in aggregated national level estimates. In particular, we characterize our sample based on individual cigarette purchase behavior in 2019 prior to the pandemic: people who do not purchase and people who purchase cigarettes in the pre-pandemic period. We further classify individuals who purchased cigarettes by their purchasing behavior as occasional smokers or regular smokers. We then estimate the impact of COVID-19 among these subgroups to understand whether the pandemic induced non-smokers into smoking, affected occasional smokers whose behavior may be particularly sensitive to social distancing measures, or shifted the habits of regular smokers who are the most likely to be addicted to cigarettes.

In the mechanisms section we note that we are not able to control for time-varying factors:

We are not able to control for changes in income (or other time-varying characteristics), but we note that in Denmark the overall unemployment rate only increased from 3.7% in 2019 to 4.6% in 2020 and salaries grew 2.5%.^{IV}

Footnote IV: Official statistics from Statistics Denmark (www.dst.dk).

9. How are cash and online purchases included in the data? Consider providing some limitations of the study to include no online and cash purchases, no control group, geographic locations, or individual characteristics, etc., if not accounted for in the model. Did the model account for these important factors?

We now include the following discussion of online and cash purchases:

Relatedly, our data largely include in person sales at grocery stores and so our results could also reflect a shift towards greater purchasing online and less in person purchasing of cigarettes. Our data do include an online supermarket, which allows us to examine the share of cigarettes purchased at the online outlet over time. The share of cigarettes purchased at the online supermarket in our data, 3.2%, is similar to the national share of online supermarket purchases, 3.6% of call grocery purchases.

In our data, we find that the share of cigarette purchases made at the online supermarket declines by 27% during the COVID-19 pandemic compared to the same period in 2019 (from 3.20% of cigarette purchases to 2.34%).

An additional limitation of our dataset is that it does not include cash purchases. However, we note that cash purchases have been steeply declining in Denmark, from 48% of purchases in 2009 to only 16% of transactions in 2019³². If the shift away from cash continued through 2020 our estimates would be upwardly biased.

10. On the readability, the study needs extensive revisions to correct several grammatical errors and incomplete sentences. For example, the sentence on page 5, paragraph 3, line 4: "...with the exception of..." is incomplete.

We apologize for overseeing this. The reference was there to complete the sentence but when reformatting the references for the journal, the name of the author got lost. We have corrected the sentence now. And have had an external proofreader review the draft for errors.

Another example is that findings in references 21 and 22 were incorrectly cited.

Thank you for the careful read. The references are now included following the Nature's formatting guide as found in: <https://www.nature.com/nature/for-authors/formatting-guide>.

Vertical dotted lines were referred to in Figure 1, but I could not find them.

We have corrected this to read "The vertical lines" (we have removed "dotted").

Please move the Materials and Methods section to the section before the Results unless it is the formatting style of the journal.

We have moved the Materials and Methods section before the Results section.

Reviewer #3 (Remarks to the Author):

This paper uses novel data from an app that allows individuals to track their grocery purchases to understand patterns of cigarette purchases and alcohol purchases in Denmark during 2020, much of which was marked by the COVID-19 pandemic, relative to 2019. In contrast to much of the emerging literature, which finds increases in tobacco and alcohol purchases and consumption during the pandemic, the paper finds marked reductions in tobacco purchases, particularly among regular smokers and no change in alcohol purchases. Those studies are not based in Denmark and so it is entirely possible the experience is just different in Denmark. That said, one is left wondering if a major part of the discrepancy may be due to the specific data analyzed. That is, as I understand the data, they do not capture purchases made outside of grocery stores. If the pandemic changed where individuals make their purchases generally or where they purchased specific items, then these data may provide a misleading picture of tobacco purchases. For example, if individuals are less likely to go to the grocery store and/or more likely to make cigarette purchases at convenience stores then actual purchases may have changed in ways not captured in the data. While the paper tries to dismiss this concern by arguing that lockdowns and social distancing did not differentially affect where people could buy tobacco or alcohol, it may have changed where people opted to make these purchases. Frequent purchases of cigarettes at smaller convenience stores, for example, could have crowded out purchases at grocery stores. Without other data – e.g., survey measures, sales data, etc. - to validate the findings, I worry that the data from this app provide a misleading picture of what is happening to tobacco and alcohol consumption in Denmark.

Thank you for this helpful feedback. In response to your comments and those of the other reviewers, we have added a new section to address these concerns. We include the following discussion of potential substitution to convenience stores:

Our data do not include purchases at convenience stores and so there may be concern that the decline in cigarette purchases we measure partially reflects a shift towards greater purchasing at corner shops and less purchasing at grocery stores. We report national administrative data on aggregate monthly sales from 2019-2020 separately for convenience stores and for grocery stores (Appendix Figure S3, monthly sales are indexed to 2015 sales). We find that during COVID-19 (compared to the same period in 2019), aggregate sales at convenience stores increased by an estimated 1.5 percent while sales at grocery stores increased by 5.5 percent. These results suggest that if anything there was a shift in consumption away from corner shops and towards supermarkets during the pandemic.

Figure S3. Convenience store and grocery store sales 2019-2020 (Index: 2015=100)

Note: The total amount of monthly sales in convenience stores and grocery stores (in Danish: “Købmand og døgnkiosker” & “Detailomsætningsindex”), indexed to average monthly sales in 2015. Data source: Statistics Denmark, www.statistikbanken.dk/DETA151 + www.statistikbanken.dk/DETA152.

Following your suggestion, we have supplemented our data with national data for Denmark and include the following discussion:

Finally, our findings align with aggregated national sales data, which show an 18% decline in cigarette sales in 2020 compared to 2019 and a 2.1% increase in alcohol sales (Appendix Figure S4). We therefore interpret our results as reflecting declines in overall cigarette purchases during the COVID-19 pandemic.

Figure S4. National Cigarette and Alcohol Sales by Year.

Panel A

Panel B

Note: Panel A shows the number of cigarettes that an average Danish adult bought in 2019 and 2020. Panel B shows the liter of pure alcohol an average Danish adult bought in 2019 and 2020. The pure alcohol measure is determined by calculating the alcohol share (in liters) of each alcohol-containing product sold. Data source: Statistics Denmark: www.statistikbanken.dk/ALKO2

Other Comments

1) The authors interpret the patterns they observe in the data as reflecting concerns about health risk, specifically concerns that tobacco consumption puts one at higher risk for adverse COVID-19 outcomes. However, the data really do not provide any measure of risk perceptions and many other factors put individuals at higher risk of adverse outcomes, including obesity. Yet, other data suggests a decline in exercise. While the paper's analysis of mechanisms provides some support for this interpretation, it really seems overly strong to attribute these patterns to concerns about the risks of smoking.

We agree that we should moderate our claims about the mechanisms. We have drawn on your comment and added the following:

Our data do not include measures of risk perceptions related to smoking and COVID-19, and so we cannot directly test for this mechanism in our population. However, two studies find evidence in the U.S. that smokers were on average more motivated to quit in response to the health risks of COVID-19.

2) The paper adds cigarette prices to some of the models and concludes that some of the changes in consumption are due to price. If the pandemic really caused a shock to demand, however, shouldn't prices have declined? Were there supply shocks due to the pandemic?

We are not aware of supply shocks due to the pandemic that affected supply during the period of our study. Prices increased during this period due to an increase in the national tax on cigarettes that went into effect partway through the pandemic. Prior work has shown theoretically and empirically that decreased demand for addictive good such as cigarettes does not necessarily lead to decreased prices, because the remaining consumers may be more inelastic. And in particular, increases in taxes can lead to price increases that are higher than the tax (see Sloan et al., 2004). We do not currently include this discussion in the text but are happy to add it if you and the editor prefer.

Sloan, Frank A., Carrie A. Mathews, and Justin G. Trogon. "Impacts of the Master Settlement Agreement on the tobacco industry." Tobacco Control 13, no. 4 (2004): 356-361.

3) The paper says that the app data capture spending on grocery purchases as well the environmental impact of the purchases. What does the environmental impact of the purchases mean?

As we now clarify, the environmental impact of purchases is the carbon footprint of purchases – i.e., the greenhouse gas emissions in CO₂ equivalents from the production process.

4) In the notes to the tables, it would make sense to clarify that the COVID-19 indicator captures weeks 11 on in any given calendar year.

We agree that the labels for the coefficients in Tables S2-S7 were confusing. We have changed the coefficient labels in our tables to clarify the different variables. We now label β_2 "weeks 11-52",

which includes both 2019 and 2020. We label β_3 “COVID-19 period” and include in the notes that “COVID-19 period is the coefficient for the interaction of Week 11-52 x Year (2020).”

5) There are many other papers on the topic now, including ones based in Europe, that should probably be added to the citations. Some examples include:

<https://academic.oup.com/eurpub/article/31/5/1076/6214519>
<https://www.sciencedirect.com/science/article/pii/S2352340921007617>
<https://www.mdpi.com/1660-4601/18/13/7128>
<https://www.ncbi.nlm.nih.gov/mc/articles/PMC7643580/>
<https://www.ncbi.nlm.nih.gov/pmc/articles/PMC7386200/>

Note that other work seems to align more with the findings from Denmark:

<https://www.sciencedirect.com/science/article/pii/S0306460321001027>
<https://academic.oup.com/ntr/article/22/9/1662/5826329>

Thank you for bringing these references to our attention. These studies are very useful, and it is important that they are included. We have updated the literature discussion to include these, and other new references. The text now reads as follows:

A rapidly emerging literature has used surveys to investigate self-reported consumption changes during the COVID-19 pandemic for diet and addictive substance use (i.e., alcohol, tobacco, drugs). The survey studies find mixed results for estimates of changes in (expected) tobacco consumption during COVID-19. Some studies report average increases in smoking¹⁶⁻²⁰, while others report average decreases^{21,25}. Within these studies, several find evidence of heterogeneity with about 15-30 percent of respondents reporting increases in smoking and a similar share reporting decreases^{19,22,23}. These studies were conducted in different countries (USA, France, Netherlands, Greece, China, South Africa, India, Italy, the UK), using a range of survey techniques (including online and phone surveys), study lengths (ranging from a week to months), and sample sizes (from a few hundred to many thousands). The only prior study in Denmark was conducted among pregnant women and finds no effects of COVID-19 on smoking among this group²⁴. All of these studies rely on self-reported measures. And, as highlighted in a recent study by the U.S. Centers for Disease Control (CDC), they may also suffer from differential response rates during the COVID-19 pandemic compared to pre-pandemic periods²⁵.

Reviewer #4 (Remarks to the Author):

The current paper examines individual level cigarette purchase data among a sample of the Danish population before versus during the COVID-19 pandemic. The authors find a decrease in cigarette purchase rates among their sample during the pandemic. They further demonstrate that this decline is largely driven by “regular smokers” and that it cannot be fully attributed to changes in price. The writing is clear and the research question, and data, is important and interesting. I do have some concerns with the presentation of measures, the discussion of potential mechanisms, and discussion/conclusion which I outline below.

I have several comments on the measures/definitions used:

1) The authors claim that their data provides evidence for declining consumption based on purchase rates. While purchase rates may serve as a proxy for consumption, this needs to be made clearer (vs. a blanket claim that consumption declined). I recommend the authors either directly say at the outset that purchase rates serve as a proxy for consumption or simply discuss purchase rates throughout the paper (and save their proposition about decreased consumption for the discussion/conclusion). Citations where purchase rates have been used as a measure of consumption would be helpful. Further, the authors should add a few sentences (perhaps in a footnote or discussion) explaining that their data does not allow them to use more common definitions of current smokers (e.g., smoked at least 100 cigarettes in lifetime, etc.).

We have revised the structured abstract to only mention purchases (and not consumption). We have also first sentence in the text in which we introduce our study (first sentence of the third paragraph of the introduction) to highlight that purchases serve as a proxy for consumption:

In this study, we examine the impact of COVID-19 on tobacco consumption as proxied by purchases among a national sample of the Danish population (N=4042). ...

We have drawn on your comment and revised footnote 1 to include both our definition of smokers and the definition of smokers that our data does not allow us to measure, which we paste below (without citations):

According to the National Institute for Health Education Risk Prevention (INPES), a regular smoker is somebody who admits to smoking at least one cigarette (or equivalent) per day. To calculate average daily cigarette purchases, we divide an individual's total cigarette purchases in 2019 by 365. Our data do not allow us to use another common survey measure of current smoking: having ever smoked 100 or more cigarettes within one's lifetime and smoking every day or some days at the time of survey.

2) In the main analyses smokers are defined as anyone who buys at least one cigarette per

day. The authors should either cite other research that has used this same measure, or provide discussion about how this differs from typical classifications of smokers and why (a more conservative approach would be to refer to these individuals as “people who purchased cigarettes” in the main analysis and then save smoker classification for the occasional/regular smokers in the secondary analyses).

We have drawn on your language and revised the introduction to emphasize purchasing behavior for our main analysis and then secondarily our occasional/regular smoker classification:

In particular, we characterize our sample based on individual cigarette purchase behavior in 2019 prior to the pandemic: people who do not purchase and people who purchase cigarettes in the pre-pandemic period. We further classify individuals who purchased cigarettes by their purchasing behavior as occasional smokers or regular smokers.

We have also revised the first sentence of the Results section describing Figure 1. We no longer use the term smokers and instead refer only to purchasing behavior:

Figure 1 plots average cigarette purchase rates (Panel A) and quantities (Panel B) by week in 2019 and 2020 among individuals who buy at least one cigarette in 2019 (24.3% of our sample).

We have similarly revised the notes to Figure 1. We no longer include the term smokers and instead use your suggested phrasing:

Panel A – Weekly average purchase rate by year; Panel B – Weekly average purchase of cigarettes by year. The two red lines indicate the lockdown period. The figure only includes individuals who purchased at least one cigarette in 2019.

3) The introduction starts out by discussing the health costs of smoking, and how these health costs have increased during the pandemic. There are two elements of this that could be improved (the latter being a larger change than the former): (1) “health costs of smoking” should be defined at its first mention on page 4—the way it is discussed feels more like health risks than health costs (which would also imply financial costs); (2) the introduction of the paper is framed such that the authors propose that increased awareness of health costs/risks is a main driver for the decrease in purchase rates. The discussion of potential mechanisms at the end of the paper, however, focuses on the increased cost of cigarettes (which I agree should be thoroughly discussed) and lists health risks last with very little discussion. This makes the front-end and back-end of the papers feel mismatched. The front-end led me to believe that health risks would not only be a potential explanation for the authors’ findings, but that the paper would provide a thorough discussion of why that may be. If the authors want to keep the framing of the paper as one about awareness of health risks, then the discussion/evidence as to why this is a relevant explanation could be strengthened.

Our aim in the introduction was to outline the ways in which the effects of COVID-19 might either increase or decrease smoking: health risks; shifts in society, including social distancing measures; and, increases in stress and declines in both physical activity and mental well-being. We have

revised the paragraph to give more equal weight to each of these. We cut several sentences discussing awareness of health risks, which we have moved to the Mechanisms section. We have also edited the first sentence of the second paragraph to read “health risks” instead of “health costs.” The revised second paragraph is pasted below (without citations):

The health risks of smoking have increased during the COVID-19 pandemic, which represents one of the greatest global shocks in modern history to both health and society. Researchers recognized early in the pandemic that tobacco smoking – which is long known to increase susceptibility to infection and activation of inflammation – is a leading risk factor for more severe COVID-19 symptoms, hospitalization, and death 6–8. The increased health risks were accompanied by unprecedented societal measures including lockdowns, social distancing, and remote working policies. There is evidence from prior work that both health shocks that increase personal risks from smoking (e.g., during pregnancy), as well as large societal shifts (e.g., smoking norms and restrictions) can drive sustained reductions in smoking 9–11. At the same time, recent studies demonstrate that the pandemic has led to reductions in physical activity, increases in stress, and declines in mental well-being 12–14, all factors commonly associated with triggering higher tobacco use 15. Taken together, the COVID-19 pandemic could increase, decrease, or have little impact on smoking, and the effects could differ across individuals.

We have also tried to create balance in the abstract. In the structured abstract, the background section discusses impacts of the pandemic that could either increase or decrease tobacco use:

The COVID-19 pandemic has increased the health costs of smoking, which is a leading risk factor for more severe COVID-19 symptoms, hospitalization, and death. The pandemic has also led to reductions in physical activity, increases in stress and declines in mental well-being, all factors commonly associated with triggering higher tobacco use.

In addition, we have cut from both the abstract and the conclusion of the structured abstract that our results are consistent with individuals responding to the increased health risks of smoking due to COVID-19.

4) Related to the above, on pg. 4 the authors mention prior research that shows that health shocks (e.g., pregnancy) decrease smoking. This is a key point and I would encourage the authors to consider highlighting this more.

Thank you for highlighting this. While we have de-emphasized our discussion of the awareness of health risks in response to your comment above, we have retained the discussion of health shocks (e.g., pregnancy) following your guidance here. Our discussion of health shocks now receives relatively more emphasis in the second paragraph of introduction (pasted above) and is also the focus of the final paragraph of the conclusion pasted below (without citations).

Whether smoking returns to pre-pandemic levels is an open question. Studies of health shocks, such as pregnancy, demonstrate large rebound effects after the temporary shock fades away: e.g., 23% had ceased smoking during pregnancy but only 8% had quit smoking

after pregnancy in Haug et al.. At the same time, the pandemic has shifted some lifestyle habits in ways that people expect to persist. If the health risk perception related to COVID-19 and smoking lasts long enough, the quitting behavior might also persist, loosening the addictive habit.

5) It would be helpful to have more information on the sample used. For instance, how do people know/sign-up for the app? Are they compensated? Is there population level data on the average % of cigarettes purchased at these locations vs. other locations that are not in the data?

We added the following information on the app users in the Method sections.

There was no recruitment for this study, as individuals were not aware that their cigarette purchase was being observed, and no compensation was involved. Users would typically find out about these apps through marketing campaigns run by the developer company. App-users can activate a profile that includes optional demographic questions and connect the app to an e-receipt system of widespread use in Denmark.

In response to your comment and the comments of other reviewers we have added a new section that includes the following discussion of locations that are not in the data:

Our data do not include purchases at convenience stores and so there may be concern that the decline in cigarette purchases we measure partially reflects a shift towards greater purchasing at corner shops and less purchasing at grocery stores. We report national administrative data on aggregate monthly sales from 2019-2020 separately for convenience stores and for grocery stores (Appendix Figure S3, monthly sales are indexed to 2015 sales). We find that during COVID-19 (compared to the same period in 2019), aggregate sales at convenience stores increased by an estimated 1.5 percent while sales at grocery stores increased by 5.5 percent. These results suggest that if anything there was a shift in consumption away from corner shops and towards supermarkets during the pandemic.

Figure S3. Convenience store and grocery store sales 2019-2020 (Index: 2015=100)

Note: The total amount of monthly sales in convenience stores and grocery stores (in Danish: “Købmand og døgnkiosker” & “Detailomsætningsindex”), indexed to average monthly sales in 2015. Data source: Statistics Denmark, www.statistikbanken.dk/DETA151 + www.statistikbanken.dk/DETA152.

6) On pg. 5 the authors briefly discuss surveys on consumption changes during the pandemic and indicate that most of this data finds that people report an increase in tobacco consumption. I have several recommendations for this section. First, the authors cite four surveys in a way that implies a comprehensive list. There have been many more surveys/studies conducted on tobacco consumption during the pandemic and the text should reflect this. Second, the location/population of the surveys cited should be mentioned.

We have revised the literature review with many more references, which also led to a more nuanced discussion about the existing findings. We also highlight that the studies are completed under very diverse settings. The text now reads as follows (reference numbers are included in the text):

A rapidly emerging literature has used surveys to investigate self-reported consumption changes during the COVID-19 pandemic for diet and addictive substance use (i.e., alcohol, tobacco, drugs). The survey studies find mixed results for estimates of changes in (expected) tobacco consumption during COVID-19. Some studies report average increases in smoking¹⁶⁻²⁰, while others report average decreases^{21,25}. Within these studies, several find evidence of heterogeneity with about 15-30 percent of respondents reporting increases in smoking and a similar share reporting decreases^{19,22,23}. These studies were conducted in different countries (USA, France, Netherlands, Greece, China, South Africa, India, Italy, the UK), using a range of survey techniques (including online and phone surveys), study lengths (ranging from a week to months), and sample sizes (from a few hundred to many thousands). The only prior study in Denmark was conducted among pregnant women and finds no effects of COVID-19 on smoking among this group²⁴. All of these studies rely on self-reported measures. And, as highlighted in a recent study by the U.S. Centers for Disease Control (CDC), they may also suffer from differential response rates during the COVID-19 pandemic compared to pre-pandemic periods²⁵.

7) Finally, as the authors suggest, many of these surveys find the opposite of what the authors find: an increase in tobacco consumption. However, the authors do not discuss why they believe their data shows a different pattern—is it that they use individual level data and these others use population level data (and if so, why would that lead to this difference?). Do they believe that there was a specific campaign that occurred in Denmark that made people more aware of the health risks than in other countries (as health risks is their main argument as to why this decline occurred), or might it have to do with price changes in Denmark that may not have occurred in other countries?

As discussed above in response to comment 6, our expanded discussion of the literature now highlights that the results of surveys find mixed results for the impact of COVID-19 on smoking. Our results differ from some of the prior studies and align with others. To the extent our results differ from other studies, it may be due to context. However, as we now highlight for the U.S., even within the same context different methods can reach different conclusions. The other primary difference of our study with prior work is that we track individual-level systematic purchasing data

for the same people prior to and during the pandemic. Our passively collected purchasing data (rather than self-reported survey measures with differential response rates) as well as our ability follow individuals over time could also help explain the differences between our results and prior studies. We have revised the introduction to highlight these features of our study.

8) The authors may consider adding a sentence or footnote on page 8 indicating that they will return to the discussion of price in the discussion section (as the results from the analyses raise questions).

That is a good idea, we have added the following footnote (Footnote III):

We discuss the effect of cigarette price in more detail in the Mechanisms section.

9) It would be nice to see the sample sizes for the different smoker groups mentioned alongside the analyses on page 8 (rather than just in the appendix).

We now include the sample sizes for each subgroup.

10) The authors should add citations to support their claim that alcohol could be a substitute for cigarettes as there is much evidence suggesting alcohol is a complement, not a substitute, for tobacco.

We have revised our discussion of alcohol that it can serve as either a complement or a substitute and have added references for both:

We also examine alcohol purchases, which may serve as a substitute or complement for cigarettes (Appendix Table S4).

Goel, Rajeev K., and Mathew J. Morey [1995] "The Interdependence of Cigarette and Liquor Demand," Southern Economic Journal, Vol. 62, No. 2, October, pp. 451-459.

Tauchmann, H., Lenz, S., Requate, T. et al. Tobacco and alcohol: complements or substitutes? Empir Econ 45, 539–566 (2013).

11) The occasional smokers show an increase in purchase quantity—this should be discussed. Further, given this, the claim on page 9 that regular smokers account “for the full decrease in cigarette consumption” is inaccurate.

We have revised this to:

Importantly, we find no evidence of larger increases in alcohol consumption among regular smokers, who drive the decrease in cigarette consumption.

And now discuss the increase in purchase quantity among occasional smokers:

At the same time, the overall decline in cigarette consumption masks heterogeneous effects of the pandemic, with evidence that occasional smokers increased their purchase quantities during COVID-19.

We also return to this point in our discussion of mechanisms:

Our findings do not provide strong evidence that the declines are driven by changes to social and work environments. We find smaller declines in purchases during the strict lockdown when there was the largest change to social and work environments, and the largest declines during the spring and summer when the economy reopened. We also find increases in purchase quantities among occasional smokers, who are likely to be those most sensitive to “social smoking.” These findings suggest social distancing measures and related restrictions are not driving the overall decline in smoking.

12) The authors should acknowledge that they discuss some, but not all, potential mechanisms (e.g., on page 10 the sentence should say something along the lines of “it could be due to a range of factors, including but not limited to...”

We have drawn on your comment and revised the first paragraph of the Mechanisms section as follows:

We consider several potential mechanisms for the decline in tobacco purchases during COVID-19. It could be due to a range of factors, including but not limited to, the increased risks from COVID-19 that smokers face, changes in social and work environments, or financial concerns.

13) In the discussion of price increases (pg. 11), the authors may also want to mention simultaneous decreases in wages/unemployment as a result of the pandemic

We now include the following brief discussion of wages/unemployment when we discuss price increases:

Relatedly, changes in financial security could affect demand for cigarettes. We are not able to control for changes in income (or other time-varying characteristics), but we note that in Denmark the overall unemployment rate only increased from 3.7% in 2019 to 4.6% in 2020 and salaries grew 2.5% .

14) The discussion on mechanisms falls a bit flat. First, as mentioned above, it feels mismatched with the intro/framing of the paper being about health risks. Second, there should be some mention that these are just a few possibilities and that there may be other potential mechanisms at play (e.g., shift in social norms during the pandemic, fear not only for one’s own health but for others, etc.). Finally, this would also be a good place to discuss

mechanisms that are specific to Denmark and could explain the discrepancy in findings between this data and the surveys that find an increase in consumption in other countries.

First, as discussed above in response to comment 3, we have revised the abstract and introduction to balance our discussion of health risks with other potential mechanisms. Second, as discussed above in response to comment 12, we have revised the discussion of mechanisms to acknowledge that other factors may be driving the effects we estimate. Finally, as discussed above in response to comments 6 and 7, our expanded literature review reveals that our results may not be specific to Denmark. In addition, we have added the following paragraph at the end of the Mechanisms section:

Many other unobserved mechanisms might explain the declining cigarette consumption, including changes in social norms or growing concerns about exposing others to passive smoking. To the extent that our findings differ from prior work, it could be due to differences in methodology as well as context. Of particular relevance in the Danish context may be the high levels of trust and confidence in the national government, which could increase responsiveness to new information about increased health risks from smoking.

15) The authors might consider adding a table to the appendix that provides more detail on the review of interventions mentioned at the bottom of pg. 11.

We draw these estimates from a recent review cited in the text and below:

*CDCP. Tobacco Control Interventions. (2017) Available at:
<https://www.cdc.gov/policy/hst/hi5/tobaccointerventions/index.html>.*

16) Pg. 12. The last sentence in the second paragraph could be broadened – not only would a continued decline protect against risks from new variants, but it could have a meaningful effect on population health and life expectancy beyond covid.

We really appreciate your suggestion. We have drawn on your language in the final sentence:

If the decline in smoking we document persists, not only could it help decrease the risks from COVID-19 as new variants emerge, but also have meaningful, longer-term benefits on population health and life expectancy beyond the pandemic.

17) The authors compare the smoking rate in their sample to that of the Danish population. If possible, it would be nice to see a comparison of the smoking rate in this sample compared to a matched demographic sample (as the sample used in this study differed from the general population in ways that may affect smoking rates, such as gender and income).

Our estimate is the result of the participating sample, and to the extent that it is not nationally representative, might be the reason for the lower share of regular smokers. We now include analyses that uses inverse probability weighting (IPW) to re-weight our sample based on the share of the different categories of smokers in the population. We report the results in Appendix Table

S8 and have pasted the table below. We now include the following discussion in the Materials and Methods section:

The proportion of regular smokers in our data is lower than in the national Danish population: an estimated 17% of Danes smoke daily. To address the non-representativeness of our sample, we include a re-weighted analysis that uses inverse probability weighting to match the shares of smokers and non-smokers observed in the national population, and the results do not change (Appendix Table S8).

Appendix Table S8. Regression on the impact of COVID-19 on cigarette quantity using Inverse Probability Weighting of the sample for the national shares of smokers and non-smokers.

IPW on cigarette quantity by smoker category										
	All	All+price	Non-smokers	Non-smokers+price	Smokers	Smokers+price	Occasionals	Occasionals+price	Regulars	Regulars+price
Spring	0.397*** (0.134)	0.193 (0.138)	0.002 (0.032)	-0.006 (0.027)	1.624*** (0.543)	0.812 (0.560)	0.333 (0.205)	0.225 (0.208)	4.237*** (1.579)	2.001 (1.640)
Summer	-0.012 (0.135)	0.084 (0.131)	-0.034 (0.027)	-0.030 (0.027)	0.055 (0.547)	0.439 (0.533)	0.077 (0.236)	0.128 (0.228)	0.012 (1.587)	1.068 (1.544)
Fall	-0.239* (0.132)	-0.050 (0.129)	-0.029 (0.026)	-0.021 (0.027)	-0.892* (0.537)	-0.142 (0.524)	0.010 (0.240)	0.110 (0.220)	-2.720* (1.547)	-0.652 (1.522)
Week 11 - 52	0.630*** (0.168)	0.587*** (0.165)	0.018 (0.024)	0.017 (0.025)	2.535*** (0.684)	2.362*** (0.669)	-0.114 (0.256)	-0.137 (0.253)	7.899*** (1.971)	7.423*** (1.930)
Year (2020)	0.215 (0.175)	0.221 (0.176)	0.097*** (0.019)	0.098*** (0.019)	0.582 (0.719)	0.605 (0.721)	0.176 (0.253)	0.179 (0.253)	1.403 (2.114)	1.467 (2.122)
COVID-19 period [^]	-0.949*** (0.215)	-0.400** (0.188)	0.059*** (0.025)	0.081** (0.037)	-4.083*** (0.875)	-1.897*** (0.764)	0.542* (0.290)	0.834*** (0.297)	-13.447*** (2.503)	-7.424*** (2.203)
Cigarettes price		-3.213*** (0.679)		-0.131 (0.138)		-12.802*** (2.736)		-1.710 (1.079)		-35.260*** (7.845)
Constant	2.784*** (0.221)	14.686*** (2.532)	0.000 (.)	0.486 (0.512)	11.449*** (0.851)	58.874*** (10.164)	1.663*** (0.143)	7.998** (3.990)	31.262*** (2.180)	161.879*** (28.859)
N	420368	420368	318136	318136	102232	102232	68432	68432	33800	33800
Clusters (individuals)	4042	4042	3059	3059	983	983	658	658	325	325

* p<0.1, ** p<0.05, *** p<0.01

[^] COVID-19 period is the coefficient for the interaction of Week 11-52 x Year (2020)

18) Another possibility that the authors may want to mention is whether people may have started purchasing cigarettes online during this time (and whether or not that is prohibited by law).

We now include the following discussion of online purchases:

Relatedly, our data largely include in person sales at grocery stores and so our results could also reflect a shift towards greater purchasing online and less in person purchasing of cigarettes. Our data do include an online supermarket, which allows us to examine the share of cigarettes purchased at the online outlet over time. The share of cigarettes purchased at the online supermarket in our data, 3.2%, is similar to the national share of online supermarket purchases, 3.6% of call grocery purchases.

In our data, we find that the share of cigarette purchases made at the online supermarket declines by 27% during the COVID-19 pandemic compared to the same period in 2019 (from 3.20% of cigarette purchases to 2.34%).

19) Much of the tobacco literature on purchase rates uses population-level data. The authors may wish to highlight the uniqueness of their individual-level data.

Thank you! We have tried to highlight our individual data in our comparison to the prior literature in the introduction.

Minor Comments:

20) The background section in the abstract should have “use” added to the end of the last sentence.

We have made this correction.

21) There should be a citation added to the first sentence after “leading causes of death globally...”

We have added the citation.

22) The following sentence is not clear: “Tobacco products are highly addictive with extremely low rates of cessation, particularly for sustained periods.”

We have clarified this to: “Tobacco products are highly addictive with extremely low rates of cessation.”

23) In several places, the authors end the sentence with a citation rather than listing out the name of the cited survey (e.g., on pg. 5 in the last paragraph they end a sentence with “with the exception of 20”). It would be more fluent to also add the name of the survey to the sentence.

We now include the author names in sentences with this structure.

I wish the authors the best of luck as they move this important research forward.

Thank you for your careful read and thoughtful comments, which we hope you agree have improved the paper.

REVIEWERS' COMMENTS:

Reviewer #1 (Remarks to the Author):

Thank you for submitting the revised version of your paper. I enjoyed reading the new version and have no further comments.

Reviewer #2 (Remarks to the Author):

I do not have further comments.

Reviewer #3 (Remarks to the Author):

The revisions are very responsive to my concerns. I have no other issues. This is a nice paper.

Reviewer #4 (Remarks to the Author):

Review of COMMSMED-21-0589

The authors have done a good job of revising this paper and have addressed many of my prior concerns. The changes to the description of the data (purchase data vs. consumption data) and the updated definition of smokers is an improvement. The added data on convenience stores (vs. grocery stores) further strengthen the paper and decrease concerns about shifts in purchase locations as an alternative explanation. I also found the introduction and literature review to be greatly improved. I have a few remaining questions and concerns, all of which can be addressed in the writing, which I outline below.

- The updated language in the abstract (purchases instead of consumption) is an improvement (as in mentioning that purchase data is used as a proxy for consumption). I would recommend following the same format throughout the paper – there are several instances (once in the introduction, and once in the second sentence in materials and methods) that say “consumption data,” which may be confusing to readers.

- Relatedly, the plain language summary says that you find a large decrease in smoking – because this paragraph does not carry the nuances outlined in the paper, you may consider changing that phrase to purchase rates.

- The manuscript states that one of the contributions is that this data allows you “to examine heterogeneous individual-level effects that may be masked in aggregated national level estimates.” I would suggest adding one sentence to this paragraph indicating that, while this appears true for this data specifically, it may not be comparable to the surveys/data listed in the paragraph directly above. As written now, it sounds like the authors are claiming their data might reveal individual-level effects that were masked in the existing US (and other country) data.

- The authors may consider expanding their discussion on their finding that purchase quantities increased among occasional smokers while purchase rates decreased. While the main effects are driven by the regular smokers, readers may be interested in the authors speculations for this pattern among occasional smokers.

- While not necessary, I believe there is likely data showing a general shift away from cash payments in 2020 as many retailers stopped taking cash during the pandemic to limit transmission – the authors may consider adding a citation to further strengthen this argument.

Minor comments:

- In the plain language summary, this sentence is not clear: “The COVID- 19 pandemic has increased the health costs of smoking, which is a leading risk factor for more severe COVID-19 symptoms, hospitalization, and death.”

- There is a typo in the following sentence (it should say “all” not “call”): “The share of cigarettes purchased at the online supermarket in our data, 3.2%, is similar to the national share of online supermarket purchases, 3.6% of call grocery purchasesVI.”

We have included our responses to Reviewer #4 below *in italics*:

Reviewer #4 (Remarks to the Author):

Review of COMMSMED-21-0589

The authors have done a good job of revising this paper and have addressed many of my prior concerns. The changes to the description of the data (purchase data vs. consumption data) and the updated definition of smokers is an improvement. The added data on convenience stores (vs. grocery stores) further strengthen the paper and decrease concerns about shifts in purchase locations as an alternative explanation. I also found the introduction and literature review to be greatly improved. I have a few remaining questions and concerns, all of which can be addressed in the writing, which I outline below.

- The updated language in the abstract (purchases instead of consumption) is an improvement (as in mentioning that purchase data is used as a proxy for consumption). I would recommend following the same format throughout the paper – there are several instances (once in the introduction, and once in the second sentence in materials and methods) that say “consumption data,” which may be confusing to readers.

We have replaced these instances of the use of “consumption” with “purchases”

- Relatedly, the plain language summary says that you find a large decrease in smoking – because this paragraph does not carry the nuances outlined in the paper, you may consider changing that phrase to purchase rates.

We have changed this to purchase rates

- The manuscript states that one of the contributions is that this data allows you “to examine heterogeneous individual-level effects that may be masked in aggregated national level estimates.” I would suggest adding one sentence to this paragraph indicating that, while this appears true for this data specifically, it may not be comparable to the surveys/data listed in the paragraph directly above. As written now, it sounds like the authors are claiming their data might reveal individual-level effects that were masked in the existing US (and other country) data.

We have added “In our context” to this sentence.

- The authors may consider expanding their discussion on their finding that purchase quantities increased among occasional smokers while purchase rates decreased. While the main effects are driven by the regular smokers, readers may be interested in the authors speculations for this pattern among occasional smokers.

We have added the following: “The decrease in purchase rates but increase in purchase quantities suggests that occasional smokers are buying cigarettes less often but making larger purchases when they do buy. This could be indicative of changes in smoking behavior in this group – for example, a shift away from smoking in social situations to smoking at home.”

- While not necessary, I believe there is likely data showing a general shift away from cash payments in 2020 as many retailers stopped taking cash during the pandemic to limit transmission – the authors may consider adding a citation to further strengthen this argument.
- Minor comments:

We have added that “stores in Denmark by law have to accept cash payments, including during the pandemic”

- In the plain language summary, this sentence is not clear: “The COVID- 19 pandemic has increased the health costs of smoking, which is a leading risk factor for more severe COVID-19 symptoms, hospitalization, and death.”

We have revised the plain language summary following the editor’s suggestion, which no longer includes this sentence.

- There is a typo in the following sentence (it should say “all” not “call”): “The share of cigarettes purchased at the online supermarket in our data, 3.2%, is similar to the national share of online supermarket purchases, 3.6% of call grocery purchasesVI.”

We have corrected the type